# MixSATGEN: Learning Graph Mixing for SAT Instance Generation

**Xinyan Chen**[†12]**, Yang Li**[†1]**, Runzhong Wang**[1]**, Junchi Yan**[*12]
[1]Department of Computer Science and Engineering & [2]Zhiyuan College
Shanghai Jiao Tong University
{moss_chen,yanglily,runzhong.wang,yanjunchi}@sjtu.edu.cn
https://github.com/Thinklab-SJTU/MixSATGEN

## Abstract

The Boolean satisfiability problem (SAT) stands as a canonical NP-complete task. In particular, the scarcity of real-world SAT instances and their usefulness for tuning SAT solvers underscore the necessity for effective and efficient ways of hard instance generation, whereas existing methods either struggle to maintain plausible hardness or suffer from limited applicability. Different from the typical construction-based methods, this paper introduces an adaptive and efficient graph interpolation approach that in place modifies the raw structure of graph-represented SAT instance by replacing it with a counterpart from another instance. Specifically, it involves a two-stage matching and mixing pipeline. The matching aims to find a correspondence map of literal nodes from two instance graphs via learned features from a matching network; while the mixing stage involves iteratively exchanging clause pairs with the highest correspondence scores until a specified replacement ratio is achieved. We further show that under our matching-mixing framework, moderate randomness can avoid hardness degradation of instances by introducing Gumbel noise. Experimental results show the superiority of our method with both resemblance in structure and hardness, and general applicability.

## 1 Introduction

Combinatorial problems are fundamental in computer science and operation research, encompassing a wide range of problems prevalent in real-world applications (Bengio et al., 2021; Li et al., 2023b; Geng et al., 2023; Li et al., 2024). Within this realm, the Boolean Satisfiability Problem (SAT) (Guo et al., 2023) holds a special significance for its versatile representational capability, which determines whether there exists an assignment of Boolean variables that satisfies a Boolean formula and in general is NP-hard. SAT has served widespread practical applications, ranging from planning (Kautz et al., 1992), verification (Clarke et al., 2001), to cryptography (Soos et al., 2009). A recent line of research has witnessed the integration of machine learning methods into SAT solving in a data-driven paradigm (Guo et al., 2023), either as promotive components for heuristics (Selsam & Bjørner, 2019; Li et al., 2022a) or in an end-to-end manner (Selsam et al., 2019; Amizadeh et al., 2018). In these methods, bipartite graph-based representation, e.g. literal-clause graph (LCG) (Biere et al., 2009), for SAT instances have been widely adopted which provides an explicit way of generating SAT instances via graph learning techniques in addition with certain heuristics (You et al., 2019; Li et al., 2023a).

Both model training for modern learning-based SAT solvers and hyperparameter tuning for traditional solvers require relevant instances to either tune the trainable model parameters or the hyperparameters. Unfortunately, SAT instances are often scarce in practice (Li et al., 2023a; Giráldez-Cru & Levy, 2017). Hence extensive efforts have been recently made to generate quality instances. Compared to hand-crafted methods (Giráldez-Cru & Levy, 2017; 2015), learning-based methods (Li et al., 2023a; You et al., 2019; Garzón et al., 2022) show the promise of automatically discerning patterns from specific data, unveiling the underlying structural priors. However, the difficulty of these works often lies in how to resemble the hardness of the given (real) instances as reference. For instance, a generation

---

[*]Correspondence author. † denotes equal contribution. This work was in part supported by NSFC (92370201, 62222607) and SJTU Trans-med Awards Research (STAR) 20210106.

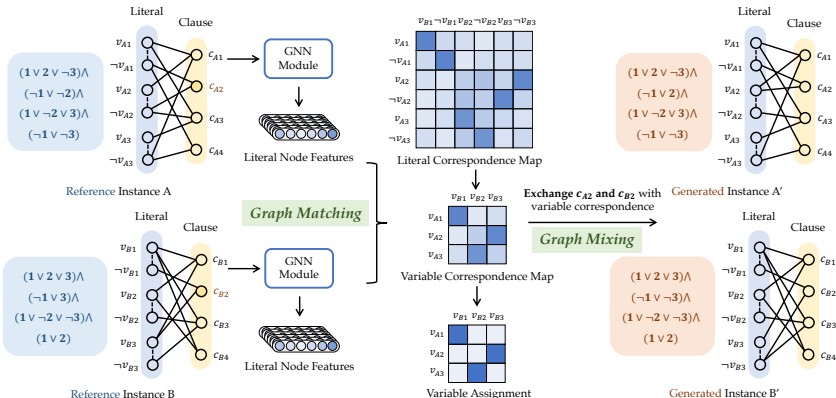

Figure 1: Methodology overview. Given a pair of bipartite graph-represented SAT instances, the matching stage predicts the correspondences of literals and variables. In the mixing stage, clause pairs are iteratively exchanged based on the mapping until a specified replacement ratio is achieved.

model trained on reference benchmarks that need hundreds of seconds to solve often generates trivial instances that can be solved in mere seconds. One notable recent progress to maintain the hardness refers to HardSATGEN (Li et al., 2023a) which proposes a multi-stage split-merge process to mimic the hardness via both learning scheme as well as heuristics in terms of logical substructures within SAT instances. The introduced heuristics may not fit arbitrary datasets and moreover, the method requires unsatisfiable reference instances which may not be available in some real-world datasets.

To overcome the challenges of previous construction methods, we resort to graph mixing (interpolation) that directly manipulates the raw structures of SAT instances in place. The resulting advantages as will also be detailed later in this paper are twofold: 1) more efficient as we do not need to generate the instance from scratch; 2) more adaptive with less human heuristics as the mixing process could be performed in a coherent optimization paradigm. We aim to discover the potential shortest editing path (Abu-Aisheh et al., 2015) from one graph instance to another and then output along this path as the generation results, as shown in Fig. 2. It provides a general and conceptually simple solution to achieve better hardness and structure resemblance, which also allows the absence of unsatisfiable instances in reference training data.

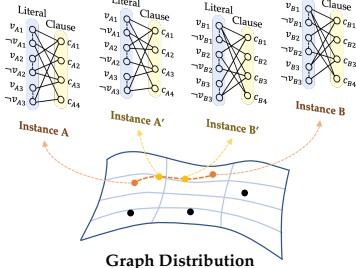

Figure 2: MixSATGEN interpolates reference instances along graph edit paths within the underlying distribution to generate new instances.

The key idea is to replace certain substructures of graph-represented SAT instances from one instance with substructures from another, and the instances are both from the reference benchmark.

Li et al. (2023a) shows that for the split-merge framework, employing similar strategies alongside similar initial structures can result in generated instances possessing similar substructures, leading to a degradation in difficulty. We posit that the same issue may be prevalent among various learning-based methods. Specifically, learning-based methods excel at identifying similarities within structures and often employ consistent learned strategies for generation. This frequently incurs the contenance of similar structures, which in turn, can easily lead to conflicts due to their close correlation, and subsequently the emergence of small unsatisfiable cores that cause hardness degradation. Based on this analysis, we argue that moderate *randomness* could help computational hardness maintenance for instance generation. This idea is supported in Table 2, by replacing certain substructures in one instance using that of another instance, random graph-level mixups of SAT instances *with a unified variable correspondence map between two instances*, can possibly retain or even elevate the solving difficulty.

Beyond random mixup between two SAT instances, we further consider learning to mix up the certain substructure given two reference instances for better structural properties preservation. The scheme of exchanging the substructures is designed by the considerations: 1) all the variables and therefore the substructures need to be matched between two instances; 2) the exchanged substructure can largely preserve the original essential patterns and the mixed instances remain within the underlying distribution of reference instances. To achieve such properties, we propose a two-stage matching and mixing pipeline dubbed MixSATGEN as a way of graph interpolation for generation as presented in

Fig. 1. The matching employs the inner products between literal node features to obtain the similarity matrix, which is subsequently post-processed using the Gumbel-Sinkhorn algorithm (Mena et al., 2018), to establish a stochastic correspondence map between the literals of two reference instances. The Gumbel noise (Jang et al., 2016) is to incorporate a controllable level of randomness to modulate the matching outcomes, avoiding the hardness degradation issue. The features are extracted by graph networks pretrained in SAT solving tasks and subsequently online optimized by the graph matching loss. The extracted node embeddings reduce the problem into a linear assignment task as solved via the Hungarian method (Kuhn, 1955), which is then enforced on the correspondence map to obtain a specific assignment of the literal nodes. In the mixing stage, clause pairs with the highest correspondence scores are iteratively exchanged until a specified replacement ratio is achieved. Experiments show MixSATGEN's comparative performance for structure and hardness resemblance as well as general applicability.

## 2 RELATED WORK

**SAT Generators.** Different from mainstream generative models (Ho et al., 2020; Li et al., 2022b; 2023c; Zhang et al., 2024b), SAT instance generation requires strong domain knowledge, and can be categorized into hand-crafted and learning-based methods. Hand-crafted methods (Giráldez-Cru & Levy, 2015; 2017) usually involve one or two structural metrics and devise algorithms to manipulate these metrics w.r.t. a reference. In general, hand-crafted generators fail to adaptively unravel specific data characteristics, merely matching partial metrics. In contrast, learning-based generators automatically capture global graph structures during training. G2SAT (You et al., 2019) and its follow-up GCN2S (Garzón et al., 2022) represent SAT instances as bipartite LCGs and develops a one-stage node split-merge framework for bipartite graph generation. Unfortunately, these works are not yet able to produce non-trivial hard instances and only generate instances that can be solved in seconds. To resolve this issue, HardSATGEN (Li et al., 2023a) takes the key substructures of real-world SAT instances e.g. the community structure and the unsatisfiable cores into consideration, and introduces a fine-grained control mechanism to mimic the hardness of the reference benchmark. One possible drawback is that it requires unsatisfiable instances for reference and relies on human intervention for post-processing. In parallel with boolean satisfiability, LinSATNet (Wang et al., 2023b) also studies the generative process that enforces linear satisfiability to the output of neural networks, and similar learning-based generators are also studied in other combinatorial problems (Geng et al., 2024).

**Graph Matching.** As a building block in our framework, we are particularly in favour of deep graph matching (Yan et al., 2020) which mainly follow two ways: 1) The quadratic methods where both node and edge affinities are explicitly modeled (Wang et al., 2021a; 2023a; Liu et al., 2022a; 2023; Lu et al., 2024); 2) The linear methods where node features embed the edge features, resulting in a linear matching problem (Yu et al., 2020; Wang et al., 2019; 2020; Jiang et al., 2022). We resort to the latter paradigm for its scalability.

**Graph Mixup.** Mixup methods (Zhang et al., 2022; Liu et al., 2022b) have been proposed for graph data augmentation. Despite methods like Wang et al. (2021b) that interpolate latent representations of graph pairs, most methods manipulate the raw graph structures. Yoo et al. (2022); Park et al. (2022) combine random subgraphs and specially selected subgraphs from different graphs to generate new graphs, respectively. (Guo & Mao, 2021; Ling et al., 2023) linearly interpolates continuous-valued adjacency matrices and feature matrices for augmentation, based on an arbitrary node-level correspondence map, respectively. Yet such mixup operations require continuous adjacency matrices and structures, which do not apply to SAT with discrete structures and varying-sized instances.

## 3 METHODOLOGY

The flow diagram of our approach as well as the section organization are presented in Fig. 3.

### 3.1 PRELIMINARIES AND FORMULATION

**SAT and its Graph Representation.** SAT is to determine the existence of a valid Boolean assignment for a Boolean formula. These formulas are constructed using Boolean variables linked by logical operators

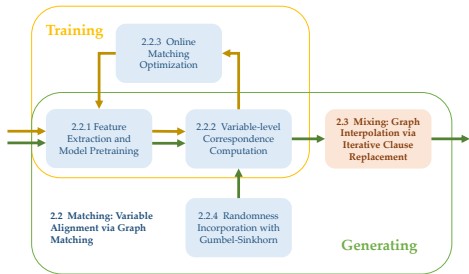

Figure 3: Flow diagram of our methodology.

like conjunction ($\wedge$), disjunction ($\vee$), and negation ($\neg$). Typically, Boolean expressions are cast in the Conjunctive Normal Form (CNF) (Tseitin, 1983) of a conjunction of clauses. Each clause represents a disjunction of literals, where a literal can signify either a variable or its negation. The graph representation of SAT compromises four basic forms (Biere et al., 2009): literal-clause graph (LCG), literal-incidence graph (LIG), variable-clause graph (VCG), and variable-incidence graph (VIG). This paper adopts LCG, which is a bipartite graph with two sets of nodes: literals and clauses. Edges within this representation signify the presence of a given literal within a specific clause.

In particular, this paper primarily adopts the LCG representation of SAT instances for its bijective nature to raw SAT instances. A bipartite LCG is formulated as $G = (\mathcal{V}, \mathcal{E})$ where $\mathcal{V} = \mathcal{V}_{lit} \cup \mathcal{V}_{cl}$ and $\mathcal{E} \subseteq \{(v_i, v_j) | v_i \in \mathcal{V}_{lit}, v_j \in \mathcal{V}_{cl}\}$. An LCG with $n$ literals and $m$ clauses maintains $\mathcal{V}_{lit} = \{l_1, \cdots, l_n\}$ and $\mathcal{V}_{cl} = \{c_1, \cdots, c_m\}$. Here $\mathcal{V}_{lit}$ and $\mathcal{V}_{cl}$ denotes the literal nodes and clause nodes, respectively. The edges can only connect literal nodes and clause nodes for the occurrence of the literal in the clause. LCG also has an equivalent representation known as Weighted VCG (WVCG). Its graph form is simplified as VCG with the negation between variables and clauses encoded into the weighted edges, where values of $1$ and $-1$ represent positive and negative literals, respectively.

**Graph Matching Objective.** Graph Matching (GM) computes the node-level assignment among two or multiple graphs based on the graph structures, which is optimal in some sense. Two-graph matching can be modeled as the Quadratic Assignment Problem (QAP), with its most general form Lawler's QAP (Leordeanu & Hebert, 2005; Yan et al., 2020) formulated as:

$$\max_{\mathbf{x}}(\mathbf{x}^\top \mathbf{K} \mathbf{x}), \quad \mathbf{X}\mathbf{1}_{n_2} = \mathbf{1}_{n_1}, \quad \mathbf{X}^\top \mathbf{1}_{n_1} \leq \mathbf{1}_{n_2}, \quad \mathbf{X} \in \{0,1\}^{n_1 \times n_2} \tag{1}$$

where $\mathbf{x} = \text{vec}(\mathbf{X})$ is column-wise vectorized $\mathbf{X} \in \{0,1\}^{n_1 \times n_2}$ which denotes the assignment matrix. Here $\mathbf{K} \in \mathbb{R}^{n_1 n_2 \times n_1 n_2}$ denotes the affinity matrix, where the diagonal elements indicate the node-to-node similarity and the off-diagonal elements carry the edge-to-edge similarity. The objective $\mathbf{x}^\top \mathbf{K} \mathbf{x}$ quantifies the similarity between nodes and edges, for maximization.

## 3.2 Matching: Variable Alignment via Graph Matching

### 3.2.1 Feature Extraction and Model Pretraining

Real-world SAT instances are often large in size, e.g. with thousands of nodes in a single LCG, which is out of the capacity of existing graph matching methods. We resort to utilizing graph networks to embed the edge information and the clause node features into the literal node features, thereby reducing the quadratic matching problem over the entire graph pairs to a linear one over literal nodes.

Following the classic NeuroSAT (Selsam et al., 2019), the graph network learns the embeddings for the literal and clause nodes through a $k$-round message passing scheme, which is formulated as:

$$\mathbf{h}_c^{(k)} = \text{LSTM}_c \left( \sum_{l \in \mathcal{N}(c)} \text{MLP}_l \left( \mathbf{h}_l^{(k-1)} \right), \mathbf{h}_c^{(k-1)} \right), \quad \mathbf{h}_l^{(k)} = \text{LSTM}_l \left( \sum_{c \in \mathcal{N}(l)} \text{MLP}_c \left( \mathbf{h}_c^{(k-1)} \right), \mathbf{h}_l^{(k-1)}, \mathbf{h}_{\neg l}^{(k-1)} \right) \tag{2}$$

where $\mathcal{N}(\cdot)$ denotes the neighboring node set, and $\mathbf{h}_l^{(k)}$ and $\mathbf{h}_c^{(k)}$ denote the $k$-th layer literal and clause embeddings, respectively. $\text{MLP}_c$ and $\text{MLP}_l$ denote two multi-layer perceptions for clauses and literals, and $\text{LSTM}_c$ and $\text{LSTM}_l$ denote the layer-norm LSTMs (Hochreiter & Schmidhuber, 1997).

For hardness-awareness, we pretrain the graph networks with a SAT solving task which predicts the binary label of each variable for a satisfying assignment. It enables the model to not only capture the graph structure but also naturally discover the inherent properties regarding the hardness. Denoting $\sigma(\cdot)$ as the Sigmoid function, the binary prediction of each variable is produced by a learned MLP, with the last-layer embeddings of each pair of literals $\mathbf{h}_l^{(N)}$ and $\mathbf{h}_{\neg l}^{(N)}$ as the input:

$$y_v = \sigma \left( \text{MLP} \left( \left[ \mathbf{h}_l^{(N)}, \mathbf{h}_{\neg l}^{(N)} \right] \right) \right) \tag{3}$$

The unsupervised training objective follows Ozolins et al. (2022) that differentially maximizes the satisfiability of a SAT instance $\phi$ over a predicted soft assignment $\mathbf{x} \in [0,1]^n$, formulated as:

$$\mathcal{L}_\phi(\mathbf{x}) = -\sum_{c \in \phi} \log \left( 1 - \prod_{i \in c^+} (1 - \mathbf{x}_i) \prod_{i \in c^-} \mathbf{x}_i \right) \tag{4}$$

where $c^+$ and $c^-$ denote the positive and negative-form variable sets. The model design and the pretraining task are specified among selections in the benchmarking work G4SATBench (Li et al., 2023d) as the most efficient ones in our proposed pipeline. The extracted literal node feature matrix is denoted as $\mathbf{H} \in \mathbb{R}^{n \times d}$, which is concatenated by $n$ $d$-dimensional literal node feature vectors.

### 3.2.2 VARIABLE-LEVEL CORRESPONDENCE COMPUTATION

Given an LCG pair $G_1, G_2$ with $2n_1, 2n_2$ literals ($n_1, n_2$ variables) as well as their literal features $\mathbf{H}_1, \mathbf{H}_2$, the similarity between the two literal sets $\mathbf{M}_l \in [0,1]^{2n_1 \times 2n_2}$ can be obtained by inner-product $\mathbf{M}_l = \mathbf{H}_1 \mathbf{H}_2^\top$. The graph structural information is already encoded in $\mathbf{H}_1, \mathbf{H}_2$ by graph networks. The variable similarity can be obtained from the literal similarity considering the two literal matching forms regarding the literal phases. For example, when matching variable $i$ and $j$, the two forms for literal matching include $\{(l_i, l_j), (\neg l_i, \neg l_j)\}$ and $\{(l_i, \neg l_j), (\neg l_i, l_j)\}$. Denoting the variable similarity matrix as $\mathbf{M}_v \in [0,1]^{n_1 \times n_2}$, the similarity between variables can be set by aggregating the scores of literal pairs within each matching form and selecting the maximum value:

$$\mathbf{M}_v[i,j] = \max\left(\mathbf{M}_l[l_i, l_j] + \mathbf{M}_l[\neg l_i, \neg l_j], \mathbf{M}_l[l_i, \neg l_j] + \mathbf{M}_l[\neg l_i, l_j]\right). \tag{5}$$

Based on the obtained variable similarity matrix $\mathbf{M}_v$, we aim to achieve an alignment $\mathbf{X} \in \{0,1\}^{n_1 \times n_2}$ between the two graphs that satisfies the stochastic constraint, formulated as:

$$\mathbf{X}\mathbf{1}_{n_2} = \mathbf{1}_{n_1}, \quad \mathbf{X}^\top \mathbf{1}_{n_1} \leq \mathbf{1}_{n_2}. \tag{6}$$

The Hungarian linear assignment solver (Kuhn, 1955) can effectively achieve such a solution satisfying the stochastic constraint in polynomial time, where the linear assignment problem is formulated as:

$$\max_{\mathbf{X}} \mathrm{tr}\left(\mathbf{X}^\top \mathbf{M}_v\right) \quad s.t. \quad \mathbf{X} \in \{0,1\}^{n_1 \times n_2}, \quad \mathbf{X}\mathbf{1}_{n_2} = \mathbf{1}_{n_1}, \quad \mathbf{X}^\top \mathbf{1}_{n_1} \leq \mathbf{1}_{n_2}, \tag{7}$$

where $\mathbf{X}$ denotes the discrete assignment matrix. To enable differentiability for further online optimization as presented in Sec. 3.2.3, we resort to the differentiable and approximate version of Hungarian, i.e. the Sinkhorn algorithm (Sinkhorn & Knopp, 1967), for a relaxed projection from the positive matrix to a stochastic matrix $\mathbf{S} \in [0,1]^{n_1 \times n_2}$ by alternatively normalizing its rows and columns. The gap between Sinkhorn and Hungarian can be controlled with Sinkhorn's hyperparameter $\tau$. A smaller $\tau$ results in a sharper output that closely approximates the discrete assignment matrix, but it leads more required iterations. Conversely, a larger $\tau$ has the opposite effect. In our pipeline,

$$\mathbf{S} = \mathrm{Sinkhorn}(\mathbf{M}_v) \quad s.t. \quad \mathbf{S}\mathbf{1}_{n_2} = \mathbf{1}_{n_1}, \quad \mathbf{S}^\top \mathbf{1}_{n_1} \leq \mathbf{1}_{n_2}, \quad \mathbf{S} \in [0,1]^{n_1 \times n_2} \tag{8}$$

serves as the node-level matching result for further online optimization. While a discrete assignment matrix for generation can still be efficiently computed through the Hungarian algorithm over $\mathbf{S}$.

### 3.2.3 ONLINE MATCHING OPTIMIZATION

For the matching purpose, the model pretrained with the solving task is subsequently online optimized for each specific instance pair to maximize the similarity between the node features and between the edge features. The optimization objective follows the classic modeling of the graph matching problem as Lawler's QAP (Leordeanu & Hebert, 2005) and replaces the discrete assignment matrix $\mathbf{X} \in \{0,1\}^{n_1 \times n_2}$ with the relaxed soft assignment matrix $\mathbf{S} \in [0,1]^{n_1 \times n_2}$ to enable differentiability:

$$J(\mathbf{s}) = -\mathbf{s}^\top \mathbf{K} \mathbf{s} \tag{9}$$

Here $\mathbf{s} = \mathrm{vec}(\mathbf{S})$ is the column-wise vectorized version of matrix $\mathbf{S}$, and $\mathbf{S}$ is the soft assignment matrix derived from Eq. (8). $\mathbf{K} \in \mathbb{R}^{n_1 n_2 \times n_1 n_2}$ denotes the affinity matrix, which contains the explicit affinity scores between node pairs and edge pairs. Minimizing $J(\mathbf{s})$ requires finding the optimal assignment matrix $\mathbf{s}$ which maximizes the matching similarity of the two graphs.

### 3.2.4 RANDOMNESS INCORPORATION WITH GUMBEL-SINKHORN

With the replacement on top of the learned variable correspondence map, similar substructures from two reference graphs can easily interact and probably lead to conflicts and consequent hardness degradation. Thus, beyond the regular matching pipeline, we introduce a tunable level of randomness to the matching results to avoid hardness degradation through Gumbel-Sinkhorn (Mena et al., 2018).

The *Gumbel trick* is known for its property to recast a difficult sampling problem as an maximization operation (Balog et al., 2017). Introducing randomness to assignments is non-trivial, while Gumbel trick on discrete assignment distributions is well studied in (Mena et al., 2018) with the following satisfying sampling property. For a random assignment $\mathbf{P}$ following the *Gumbel-Matching* distribution (Mena et al., 2018) with parameter $\mathbf{M}$, i.e. $\mathbf{P} \sim \mathcal{G}.\mathcal{M}.(\mathbf{M})$, it can be verified that $H(\mathbf{M} + \mathbf{\Gamma}) \sim \mathcal{G}.\mathcal{M}.(\mathbf{M})$, where $\mathbf{\Gamma}$ denotes a Gumbel noise matrix. Here $\mathbf{M}$ can be viewed as the

| (a) Global confidence | (b) Local confidence for | (c) Global confidence for | (d) Clause Replacement |
| for every clause in $G_1$ | related clauses in $G_2$ | remained clauses in $G_2$ | for Graph mixup |

Figure 4: Single clause replacement iteration. Node color indicates confidence, line color indicates literal phases and weighted edges, and the dotted node signifies an outlier. (a): the clause for replacing in $G_1$ is determined based on global confidence. (b) and (c): tracking the most matching clause in $G_2$ referring to the local and global confidence. (d): the two selected clauses are exchanged.

similarity matrix, and $H(\cdot)$ denotes the matching operator providing the best matching results, which can be achieved by the Hungarian algorithm for the linear matching case.

To sample from the Gumbel-Matching distribution $\mathcal{G}.\mathcal{M}.(\mathbf{M}_v)$ with randomness rather than directly use the certain solution of the linear matching problem, we extend the correspondence computation process in Sec. 3.2.2 by replacing the raw Sinkhorn algorithm with the Gumbel-Sinkhorn in Eq. (8):

$$\mathbf{S} = \text{Sinkhorn}\left(\left(\mathbf{M}_v + \lambda\mathbf{\Gamma}\right)/\tau\right) \tag{10}$$

where $\lambda$ denotes the factor controlling the noise intensity, and $\tau$ controls the significance of parameter $\mathbf{M}$ on the entire distribution. A larger $\tau$ diminishes the impact of the initial matrix and introduces increased randomness, and vice versa. Then $H(\mathbf{M})$ can be viewed as samples from the assignment distribution centered with the parameter $\mathbf{M}$ and variated with the randomness controlled by $\lambda$ and $\tau$.

### 3.3 MIXING: GRAPH INTERPOLATION VIA ITERATIVE CLAUSE REPLACEMENT

Since the similarity of different pairs can vary much, not all pairs can yield quality matching results. To this end, we introduce an entropy-based filter to first select potentially valuable pairs for mixing, which assesses and filters pairs using the row-wise entropy of the soft alignment matrix $\mathbf{S} \in \mathbb{R}^{n_1 \times n_2}$:

$$h = -\sum_{i=0}^{n_1}\sum_{j=0}^{n_2} s'_{ij} \log s'_{ij} \quad \text{where} \quad s'_{ij} = \frac{s_{ij}}{\sum_k s_{ik}} \tag{11}$$

which implies the sharpness of the soft alignment matrix between $G_1$ and $G_2$. A smaller value of $h$ denotes more distinct matching relations, leading to a potentially better pair for mixing.

Subsequently, given a selected LCG pair $G_1, G_2$ with $2n_1, 2n_2$ literals ($n_1, n_2$ variables) from the filter, as well as the discrete variable assignment matrix $\mathbf{P}$ obtained from the Hungarian algorithm, we proceed to perform iterative clause-level replacement for graph mixing. Based on the variable assignment, the literal assignment can be derived simply by ascertaining the matching of literal phases. This determination can be obtained by comparing $(\mathbf{M}_l[l_i, l_j] + \mathbf{M}_l[\neg l_i, \neg l_j])$ and $(\mathbf{M}_l[l_i, \neg l_j] + \mathbf{M}_l[l_i, \neg l_j])$ in Eq. (5), while the corresponding confidence of each matched literal pair can be readily received from the relevant entry of the Sinkhorn output matrix $\mathbf{S}$. Note that the two reference graphs often differ in size, which could result in unmatched outlier nodes within the larger graph.

The mixing process in Fig. 4, commences with computing the global confidence for the clauses in $G_1$. Each clause $c$ is assigned a confidence score derived as the sum of the confidences associated with its constituent variables. These confidence scores guide the ordering sequence in which clauses are selected for replacement. Once $c_1$ in $G_1$ is identified, its contained variables and their corresponding matches in $G_2$ are traced to locate the group of clauses in $G_2$ linked to the matching variables. A local confidence score is assigned to each related clause in $G_2$, based on the sum of traced variables linked to it. By sorting the related clauses, we can select a more refined group by excluding low-scoring clauses. Another global confidence is computed for the remaining group of clauses in $G_2$, and the clause $c_2$ with the highest score is determined for replacement. In the replacement stage, we remove all original edges associated with $c_1$ and inherit $c_2$'s match-replaced variable connection. During replacement, if $c_2$ contains an outlier variable that cannot be matched to any variable in $G_1$, a new variable is created in $G_1$, and the correspondence relation between the two variables is stored in the map. The clause-level mixing iterations continue until reaching the mixing ratio of clauses.

## 4 EXPERIMENTS

### 4.1 EXPERIMENTAL SETUP

**Datasets.** The real-world SAT instances are collected from SATLIB benchmark library (Hoos & Stützle, 2000) and SAT Competition 2021 (Balyo et al., 2021). We separate the dataset into two categories, EASY and HARD, by the computational hardness, to assess the performance of MixSATGEN across varying levels of computational hardness. The instances are preprocessed by removing duplicate clauses. As a result, the processed instances encompass variable counts ranging from 82 to 450 and clause counts spanning from 327 to 5424. Half of the selected instances, i.e. instances in the EASY dataset, exhibit low computational hardness, as they could be solved by CaDiCaL (Fleury & Heisinger, 2020) in less than a second. The remaining instances in the HARD dataset maintain a substantial solving time regarding the same SAT solver. The precise solving times for each formula are provided as reference in Table 2. Each dataset includes 4 satisfiable formulas ($a$-$d$) and 4 unsatisfiable formulas ($e$-$h$) to ensure a balanced representation of satisfiability.

**Baselines.** We involve both hand-crafted generators and learning-based generators in our comparative analysis of MixSATGEN. The hand-crafted generators include the Community Attachment (CA) model (Giráldez-Cru & Levy, 2015) and the Popularity-Similarity (PS) model (Giráldez-Cru & Levy, 2017). For learning-based SAT generators, we include G2SAT (You et al., 2019) and GCN2S (Garzón et al., 2022), both of which employ a one-stage node split-merge framework with different network backbones. The more recently proposed HardSATGEN (Li et al., 2023a) is also compared, which employs an improved multi-stage split-merge pipeline, excelling in maintaining computational hardness, albeit limited to the unsatisfiable benchmark. A fully random mixing scheme is also established within MixSATGEN's framework based on random variable assignment maps.

**MixSATGEN Settings.** We first finetune the pretrained model with 0.0001 learning rate, 200 epoches and 4 iterations of message passing (Selsam et al., 2019). For the filtering process mentioned in Sec.3.3, we select the top 3 pairs with the lowest row-wise entropy out of the pairings within the dataset for each instance. We set the mixing ratio at 5% in the main experiments. We adjust $\lambda$ to control the randomness level, while the temperature parameter $\tau$ is set as 1 in main experiments. The graph matching methods are implemented through Pygmtools (Wang et al., 2024).

### 4.2 PERFORMANCE ON GRAPH STRUCTURES

The benchmarks originating from practical applications encode distinct problems with different structural distributions. The generated formulas are desired to exhibit similar structural properties to those in the training dataset. We follow the previous studies (Ansótegui et al., 2009; Newsham et al., 2014; You et al., 2019; Li et al., 2023a) to evaluate the structural properties by graph statistics. The metrics include modularity Newman & Girvan (2004) in VIG, VCG, LIG, LCG, averaging clustering coefficient Newman (2001) in VIG. Table 1 shows that both hand-crafted methods achieve a certain level of similarity in some metrics; however, they fall short in capturing others. In contrast, the learning-based methods exhibit a more comprehensive emulation of the overall structure. Note that HardSATGEN excels in reconstruction for some metrics, particularly on the hard dataset. However, it is limited to unsatisfiable training instances, which limits its applicability. Since the other methods are aligned with the overall datasets, we do not directly compare HardSATGEN to them but merely present its statistics. The empirical results demonstrate MixSATGEN's competitive structural retention across most metrics. Furthermore, the variation trend highlights the effectiveness of Gumbel noise introduction, ranging MixSATGEN ($\lambda = 0$) that signifies a completely deterministic matching to MixSATGEN (random) with a completely random assignment. Generally, we observe that the less randomness added to MixSATGEN, the more precise it becomes in capturing structural properties.

### 4.3 PERFORMANCE ON MAINTAINING COMPUTATIONAL HARDNESS

As expected in the ten key challenges in propositional reasoning and search (Kautz et al., 1997), the generated formulas need to closely replicate the computational properties of the references. Following Li et al. (2023a), we employ the direct SAT solver runtime for assessing the hardness. We utilize CaDiCaL (Fleury & Heisinger, 2020) as the test solver and leverage the runlim tool[1] to measure the solving time for each individual instance. The maximum solving time limitation for an instance is set as 2,500 seconds. Given the potential significant variation in solving time

---

[1]http://fmv.jku.at/runlim/

Table 1: Evaluation on graph structural properties. The relative errors are in brackets. Best resemblance in **bold**. *HardSATGEN only compares to unsatisfiable reference instances.

| Dataset | Method | VIG | | VCG | LIG | LCG |
|---|---|---|---|---|---|---|
| | | Clustering | Modularity | Modularity | Modularity | Modularity |
| | Reference | 0.47±0.08 | 0.55±0.08 | 0.75±0.06 | 0.68±0.19 | 0.67±0.06 |
| | Reference (UNSAT) | 0.51±0.09 | 0.49±0.03 | 0.71±0.08 | 0.54±0.02 | 0.64±0.07 |
| EASY | CA (Giráldez-Cru & Levy, 2015) | 0.31±0.07 (33.36%) | 0.51±0.09 (7.40%) | 0.63±0.06 (16.53%) | 0.54±0.08 (20.61%) | 0.53±0.04 (21.25%) |
| | PS (Giráldez-Cru & Levy, 2017) | 0.61±0.12 (30.54%) | 0.05±0.01 (91.21%) | 0.24±0.01 (68.42%) | 0.09±0.01 (86.18%) | 0.26±0.02 (61.46%) |
| | G2SAT (You et al., 2019) | 0.55±0.15 (16.53%) | 0.61±0.16 (10.56%) | 0.82±0.04 (9.06%) | 0.82±0.19 (19.86%) | 0.72±0.06 (7.53%) |
| | GCN2S (Garzón et al., 2022) | 0.34±0.07 (27.22%) | 0.44±0.04 (19.62%) | 0.68±0.04 (9.84%) | **0.68±0.07 (1.00%)** | 0.63±0.04 (6.45%) |
| | HardSATGEN (Li et al., 2023a) | 0.41±0.10 (20.35%) | 0.43±0.06 (12.94%) | 0.63±0.03 (11.59%) | 0.54±0.04 (0.22%) | 0.61±0.04 (5.28%) |
| | MixSATGEN ($\lambda=0$) | **0.43 ± 0.06 (8.49%)** | **0.54 ± 0.10 (1.01%)** | 0.75 ± 0.03 (0.32%) | 0.68 ± 0.16 (1.21%) | **0.68 ± 0.04 (1.50%)** |
| | MixSATGEN ($\lambda=0.1, \tau=1$) | 0.42 ± 0.06 (10.38%) | 0.54 ± 0.10 (2.26%) | 0.75 ± 0.03 (0.26%) | 0.68 ± 0.16 (1.37%) | 0.68 ± 0.04 (1.29%) |
| | MixSATGEN ($\lambda=0.5, \tau=1$) | 0.41 ± 0.07 (12.17%) | 0.54 ± 0.10 (2.07%) | **0.75 ± 0.03 (0.09%)** | 0.67 ± 0.15 (1.76%) | 0.69 ± 0.04 (2.08%) |
| | MixSATGEN ($\lambda=1, \tau=1$) | 0.40 ± 0.06 (14.55%) | 0.53 ± 0.09 (3.86%) | 0.74 ± 0.03 (0.98%) | 0.67 ± 0.15 (2.66%) | 0.68 ± 0.04 (1.76%) |
| | MixSATGEN (random) | 0.35 ± 0.11 (25.63%) | 0.51 ± 0.09 (6.32%) | 0.73 ± 0.03 (2.65%) | 0.64 ± 0.14 (6.17%) | 0.68 ± 0.04 (1.54%) |
| | Reference | 0.41±0.17 | 0.59±0.18 | 0.74±0.13 | 0.67±0.16 | 0.72±0.12 |
| | Reference (UNSAT) | 0.27±0.09 | 0.45±0.19 | 0.66±0.14 | 0.62±0.23 | 0.67±0.15 |
| HARD | CA (Giráldez-Cru & Levy, 2015) | 0.27±0.06 (52.88%) | 0.45±0.15 (29.63%) | 0.69±0.14 (8.44%) | 0.56±0.16 (20.63%) | 0.60±0.12 (19.89%) |
| | PS (Giráldez-Cru & Levy, 2017) | 0.53±0.25 (27.79%) | 0.06±0.04 (89.12%) | 0.25±0.04 (66.57%) | 0.12±0.06 (82.66%) | 0.26±0.04 (63.47%) |
| | G2SAT (You et al., 2019) | 0.56±0.20 (36.71%) | 0.74±0.18 (26.93%) | 0.81±0.14 (8.26%) | 0.87±0.10 (30.18%) | 0.76±0.08 (6.48%) |
| | GCN2S (Garzón et al., 2022) | 0.38±0.19 (6.97%) | 0.65±0.20 (11.40%) | 0.76±0.13 (2.48%) | 0.83±0.10 (23.83%) | **0.72±0.07 (0.05%)** |
| | HardSATGEN (Li et al., 2023a) | 0.28±0.10 (4.66%) | 0.45±0.16 (0.73%) | 0.66±0.13 (0.09%) | 0.60±0.20 (2.61%) | 0.65±0.12 (3.12%) |
| | MixSATGEN | **0.39 ± 0.15 (4.90%)** | **0.55 ± 0.16 (6.84%)** | **0.74 ± 0.13 (0.92%)** | **0.66 ± 0.14 (2.04%)** | **0.71 ± 0.12 (1.30%)** |
| | MixSATGEN ($\lambda=0.1, \tau=1$) | 0.39 ± 0.15 (5.82%) | 0.54 ± 0.16 (7.74%) | 0.74 ± 0.13 (1.19%) | 0.65 ± 0.14 (2.50%) | 0.70 ± 0.12 (1.74%) |
| | MixSATGEN ($\lambda=0.5, \tau=1$) | 0.38 ± 0.14 (7.34%) | 0.54 ± 0.15 (7.93%) | 0.73 ± 0.13 (1.36%) | 0.65 ± 0.14 (3.55%) | 0.70 ± 0.12 (2.11%) |
| | MixSATGEN ($\lambda=1, \tau=1$) | 0.38 ± 0.14 (7.74%) | 0.53 ± 0.15 (8.86%) | 0.73 ± 0.13 (1.69%) | 0.64 ± 0.13 (4.13%) | 0.70 ± 0.12 (2.42%) |
| | MixSATGEN (random) | 0.36±0.08 (12.26%) | 0.47±0.09 (19.98%) | 0.76±0.11 (2.05%) | 0.60±0.10 (10.58%) | 0.73±0.11 (2.06%) |

Table 2: Solver runtime evaluation. Standard deviations are in brackets. Best resemblance in **bold**.

| Method | easy (average) | hard-a | hard-b | hard-c | hard-d | hard-e | hard-f | hard-g | hard-h |
|---|---|---|---|---|---|---|---|---|---|
| Reference | 0.01 | 254.22 | 1382.71 | 1204.43 | 852.85 | 2629.03 | 65.28 | 85.26 | 1532.14 |
| CA (Giráldez-Cru & Levy, 2015) | 0.01 (±0.01) | 0.01 (±0.00) | 0.01 (±0.00) | 0.01 (±0.00) | 0.01 (±0.00) | 0.01 (±0.00) | 0.05 (±0.02) | 0.01 (±0.00) | 0.01 (±0.00) |
| PS (Giráldez-Cru & Levy, 2017) | 0.01 (±0.01) | 0.41 (±0.12) | 0.41 (±0.12) | 0.01 (±0.00) | 0.01 (±0.00) | 0.01 (±0.00) | 0.01 (±0.00) | 0.01 (±0.00) | 0.01 (±0.00) |
| G2SAT (You et al., 2019) | 0.01 (±0.01) | 0.01 (±0.00) | 0.01 (±0.00) | 0.01 (±0.00) | 0.01 (±0.00) | 0.01 (±0.00) | 0.01 (±0.00) | 0.01 (±0.00) | 0.01 (±0.00) |
| GCN2S (Garzón et al., 2022) | 0.01 (±0.01) | 857.47 (±1161.46) | 102.12 (±140.05) | 0.01 (±0.00) | 0.01 (±0.00) | 0.01 (±0.00) | 0.01 (±0.00) | 0.01 (±0.00) | 0.01 (±0.00) |
| HardSATGEN (Li et al., 2023a) | 0.01 (±0.01) | N/A | N/A | N/A | N/A | 2267.95 (±90.59) | 64.18 (±3.14) | 75.29 (±5.01) | 1038.35 (±98.40) |
| MixSATGEN ($\lambda=0$) | 0.01 (±0.01) | 869.14 (±1224.09) | 358.70 (±498.90) | **889.78 (±1208.95)** | 42.48 (±85.98) | 4.75 (±3.71) | 0.01 (±0.01) | 0.01 (±0.01) | 0.01 (±0.01) |
| MixSATGEN ($\lambda=0.1, \tau=1$) | 0.01 (±0.01) | 841.34 (±1244.05) | 387.27 (±480.80) | 837.49 (±1246.90) | 67.40 (±137.58) | 15.92 (±17.75) | 0.01 (±0.00) | 0.01 (±0.00) | 0.01 (±0.00) |
| MixSATGEN ($\lambda=0.5, \tau=1$) | 0.01 (±0.01) | 567.64 (±831.01) | **470.22 (±687.26)** | 775.46 (±1057.41) | 42.65 (±60.14) | 167.70 (±249.33) | 0.01 (±0.00) | 0.01 (±0.00) | 0.01 (±0.00) |
| MixSATGEN ($\lambda=1.0, \tau=1$) | 0.01 (±0.01) | **308.67 (±448.11)** | 413.88 (±344.23) | 208.40 (±308.63) | 65.05 (±93.46) | 9.64 (±10.26) | 0.01 (±0.00) | 0.01 (±0.00) | 0.01 (±0.00) |
| MixSATGEN (random) | 0.07 (±0.41) | 623.13 (±972.90) | 418.73 (±798.925) | 478.64 (±899.21) | **174.41 (±441.21)** | 5.19 (±6.97) | 0.01 (±0.00) | 0.01 (±0.00) | 0.01 (±0.00) |

among different reference instances, we report results for each reference instance. Table 1 shows the superiority of MixSATGEN in generating promising hard instances meanwhile exhibiting general applicability. Almost all baselines experience a notable degradation regarding computational hardness. HardSATGEN excels in hardness maintenance, yet is limited for unsatisfiable benchmarks (Li et al., 2023a). The results of varying $\lambda$ show its impact on controlling the introduction of randomness. Increasing $\lambda$ tends to push the method towards a direction of complete randomness. However, due to the inherent variability introduced by randomness, not all examples conform precisely to this trend.

## 4.4 SOLVER HYPERPARAMETER TUNING BASED ON GENERATED INSTANCES.

SAT solvers with default hyperparameters perform well on general benchmarks. While in practice, with sufficient data in hand, SAT solvers can be further tuned for specific data sources. Table 3 presents the hyperparameter tuning results on the SOTA solver Kissat (Fleury & Heisinger, 2020). We adopt a Bayesian optimizer HEBO (Cowen-Rivers et al., 2022) to tune the solver. The tuned hyperparameters include restart interval, reduce interval, and decay, which pose impacts on the frequency of restarting, frequency of learned clause reduction, and per mille scores decay respectively. The tuning range of these three hyperparameters are [1, 1e2], [1e2, 1e4], and [1, 2e2] respectively. For each method, the Bayesian optimizer tries 200 sets of hyperparameters and we select the top 5 hyperparameters to evaluate on test instances. The best runtime results are presented in the table. Due to the costly Bayesian optimization process, we select instances with runtimes under 500 seconds for experiments. Note that in Table 2, MixSATGEN reproduces computational hardness well on satisfiable instances, while HardSATGEN (Li et al., 2023a) can only learn from unsatisfiable instances. Table 3 shows the matched performance where MixSATGEN and HardSATGEN perform the best for satisfiable and unsatisfiable instances respectively. Thus, we also consider simultaneously adopting MixSATGEN learned from satisfiable instances and HardSATGEN learned from unsatisfiable instances for tuning, which presents the best overall performance. In general, the generated instances by the proposed methods bring a significant solver runtime reduction of over 50% to the solver.

Table 3: Results of solver hyperparameter tuning for specific data. $T_g$: average solving runtime on generated instances; $T_t$: average runtime on test instances; $T_{hard\text{-}i}$: runtime of instance hard-$i$. Time is in seconds. Best performance for satisfiable, unsatisfiable, and overall data are marked in **bold**.

| Method | Hyperparameters | $T_g$ | Average | Satisfiable | | | Unsatisfiable | | |
|---|---|---|---|---|---|---|---|---|---|
| | | | $T_t$ | $T_{hard\text{-}a}$ | $T_{hard\text{-}d}$ | Average | $T_{hard\text{-}f}$ | $T_{hard\text{-}g}$ | Average |
| Default | (1, 1000, 50) | N/A | 130.96 | 155.60 | 254.14 | 204.87 | 33.09 | 81.00 | 57.05 |
| CA (Giráldez-Cru & Levy, 2015) | (5, 8478, 32) | 0.03 | 334.64 (-155.5%) | 84.52 (+45.7%) | 1086.08 (-327.4%) | 585.30 (-185.7%) | 55.9 (-69.0%) | 112.04 (-38.3%) | 83.98 (-47.2%) |
| PS (Giráldez-Cru & Levy, 2017) | (35, 3598, 16) | 0.03 | 207.965 (-58.8%) | 36.29 (+76.7%) | 662.47 (-160.7%) | 349.38 (-70.5%) | 54.54 (-64.8%) | 78.56 (+3.0%) | 66.55 (-16.7%) |
| G2SAT (You et al., 2019) | (87, 1276, 28) | 0.02 | 307.68 (-135.0%) | 631.93 (-306.1%) | 456.39 (-79.6%) | 544.16 (-165.6%) | 67.21 (-103.1%) | 75.19 (7.2%) | 71.20 (-24.8%) |
| GCN2S (Garzón et al., 2022) | (18, 1585, 98) | 0.73 | 240.75 (-83.8%) | 325.62 (-109.3%) | 503.32 (-98.1%) | 414.47 (-102.3%) | 51.74 (-56.4%) | 82.32 (-1.6%) | 67.03 (-17.5%) |
| HardSATGEN (Li et al., 2023a) | (2, 985, 13) | 49.80 | 197.72 (-51.0%) | 25.35 (+83.7%) | 665.29 (-161.8%) | 345.32 (-68.6%) | 30.12 (+9.0%) | 70.12 (+13.4%) | **50.12 (+12.1%)** |
| MixSATGEN | (67, 2607, 98) | 31.74 | 64.54 (+50.7%) | 46.16 (+70.3%) | 22.82 (+91.0%) | **34.49 (+83.2%)** | 66.48 (-100.9%) | 122.70 (-51.5%) | 94.59 (-65.8%) |
| MixSATGEN + HardSATGEN | (1, 791, 1) | 45.96 | **45.79 (+65.0%)** | 26.20 (+83.2%) | 44.25 (+82.6%) | 35.23 (+82.8%) | 34.20 (-3.4%) | 78.52 (+3.1%) | 56.36 (+1.2%) |

Table 4: Solver runtime with varying mixing ratio. Table 5: Solver runtime with varying randomness.

| Ratio | Method | pair b-a | pair a-d | pair b-d | pair d-c |
|---|---|---|---|---|---|
| 5% | MixSATGEN | 1023.25±4.69 | 23.72±2.29 | 138.11±18.68 | 199.63±190.70 |
| | Random | 33.47±38.53 | 1595.52±1272.20 | 1116.42±1203.59 | 695.24±799.58 |
| 10% | MixSATGEN | 1042.53±4.78 | 10.57±0.31 | 0.92±0.06 | 3.56±1.69 |
| | Random | 14.50±18.11 | 591.44±737.09 | 577.72±729.90 | 1054.32±908.35 |
| 25% | MixSATGEN | 1045.76±6.55 | 0.41±0.02 | 0.09±0.02 | 0.31±0.04 |
| | Random | 0.40±0.33 | 20.65±22.58 | 19.98±21.05 | 8.56±6.61 |
| 50% | MixSATGEN | 1047.00±3.95 | 0.01±0.00 | 0.02±0.01 | 0.13±0.10 |
| | Random | 9.33±15.06 | 0.01±0.00 | 0.01±0.00 | 0.12±0.10 |

| Reference | pair a-c | pair b-d | pair d-c | pair e-f |
|---|---|---|---|---|
| | 254.22 | 1382.71 | 852.85 | 2629.03 |
| $\lambda = 0$ | 0.09±0.01 | 52.73±3.87 | 117.83±129.39 | 0.31±0.01 |
| $\lambda = 0.1$ | 0.29±0.01 | 138.11±18.68 | 199.63±190.70 | 33.95±0.03 |
| $\lambda = 0.5$ | 0.19±0.02 | 23.14±0.74 | 122.32±12.33 | 500.12±4.60 |
| $\lambda = 1$ | 3.59±0.06 | 442.11±3.53 | 188.93±18.49 | 21.42±2.41 |
| $\lambda = 10$ | 39.88±6.91 | 332.05±25.57 | 1530.78±838.56 | 2.05±1.80 |
| $\lambda = 100$ | 130.32±63.88 | 472.43±13.31 | 1429.93±308.41 | 0.69±0.68 |
| $\lambda = \infty$ | 215.73±120.95 | 1116.42±1203.59 | 695.24±799.58 | 11.49±9.12 |

## 4.5 ABLATION STUDY

**Mixing Ratio.** Due to the large mixing ratio and different example features, it is more meaningful to analyze a single sample than mean performance. We select four of the most representative instance pairs on the HARD dataset for display. Table 4 shows the runtime variations during the replacement ratio increment for the selected instance pairs. MixSATGEN is evaluated under the setting of $\lambda = 0.1, \tau = 1$, with a comparison to that of a completely random setting. For pairs $b$-$a$ where MixSATGEN outperforms the random setting, the assignment matrix obtained is quite sharp with certainty, resulting in promising hardness maintenance when the replacement ratio increases. While for pairs $a$-$d$, $b$-$d$ and $d$-$c$ where the random setting surpasses MixSATGEN, the random side experiences some degradation, yet still maintains a plausible difficulty level at 25% replacement ratio. This provides an opportunity to enhance the difficulty generated by MixSATGEN at the same ratio, as the random setting introduces more computational hardness compared to MixSATGEN.

**Randomness.** Note Table 2 presents the average solver runtime of all pairs for each instance. However, as the hardness of some deterministic generated formulas ($\lambda = 0$) is better maintained than that of the completely random ones, the average solver runtime cannot effectively reflect the improvement of hardness led by introducing randomness. Table 5 presents the results on instance pairs where MixSATGEN exhibits some degree of hardness degradation to straightforwardly reiterate the effect of randomness on the computational hardness of generating formulas. We gradually increases the $\lambda$ value to enhance the randomness. As seen, when the randomness increases, the hardness of the initially degraded pairs is recovered to a certain extent compared with the reference. Note although pair $e$-$f$ fails to maintain hardness on both deterministic ($\lambda = 0$) and purely random formulas, it gains an amount of hardness when partial randomness is introduced ($\lambda = 0.5$).

## 5 CONCLUSION AND OUTLOOK

This paper presents an interpolation-based matching and mixing approach namely MixSATGEN that modifies the raw structure of graph-represented SAT instances by exchanging their substructures. The matching process efficiently provides the informative node-level correspondence for subsequent substructure mixing. Then we introduce controlled randomness via Gumbel noise to the matching matrices to prevent hardness degradation. MixSATGEN less relies on heuristics and human interventions, compared with HardSATGEN (Li et al., 2023a) and can handle satisfiable benchmarks that HardSATGEN cannot learn from. Empirical results show its superiority in structure resemblance and hard instance generation as well as general applicability. One future work is the study on combining strong learning-based solvers (Li et al., 2023b; Zhang et al., 2024a) with instance generation scheme e.g. in the sense of training data augmentation which has been common in vision (Zhu et al., 2020). It is also possible to incorporate multiple graphs for matching (Yan et al., 2014; Jiang et al., 2021).

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

# APPENDIX

## A  SUPPLEMENTARY EXPERIMENTS

All the experiments are performed on a single GPU of GeForce RTX 3090. The affinity matrix $\mathbf{K}$ is calculated on an AMD Ryzen Threadripper 3970X 32-core CPU with 128GB memory.

### A.1  MORE RESULTS OF SOLVER RUNTIME

Table 6 and Table 7 show the evaluation of solvers Kissat (Fleury & Heisinger, 2020) and SBVA-Cadical (Haberlandt et al., 2023)[2] runtime over the generated formulas as well as the original benchmarks. We set the runtime limit for Kissat at 5000 seconds, while for SBVA-Cadical, no runtime limit is imposed due to its slower execution on the Hard dataset. As seen, MixSATGEN perform consistently regarding the computational hardness across different solvers.

Table 6: Kissat runtime evaluation. Standard deviations are in brackets. Best resemblance in **bold**.

| Method | hard-$a$ | hard-$b$ | hard-$c$ | hard-$d$ | hard-$e$ | hard-$f$ | hard-$g$ | hard-$h$ |
|---|---|---|---|---|---|---|---|---|
| Ground Truth | 155.60 | 1216.28 | 1478.99 | 254.14 | 2033.51 | 33.09 | 81.00 | 1898.28 |
| CA (Giráldez-Cru & Levy, 2015) | 0.01 ($\pm$0.00) | 0.01 ($\pm$0.00) | 0.01 ($\pm$0.00) | 0.01 ($\pm$0.00) | 0.01 ($\pm$0.00) | 0.04 ($\pm$0.02) | 0.01 ($\pm$0.00) | 0.01 ($\pm$0.00) |
| PS (Giráldez-Cru & Levy, 2017) | 0.07 ($\pm$0.01) | 0.07 ($\pm$0.02) | 0.01 ($\pm$0.00) | 0.01 ($\pm$0.00) | 0.01 ($\pm$0.00) | 0.01 ($\pm$0.00) | 0.01 ($\pm$0.00) | 0.01 ($\pm$0.00) |
| G2SAT (You et al., 2019) | 0.01 ($\pm$0.00) | 0.01 ($\pm$0.00) | 0.01 ($\pm$0.00) | 0.01 ($\pm$0.00) | 0.01 ($\pm$0.00) | 0.01 ($\pm$0.00) | 0.01 ($\pm$0.00) | 0.01 ($\pm$0.00) |
| GCN2S (Garzón et al., 2022) | 1686.27 ($\pm$2343.27) | 16.18 ($\pm$17.42) | 0.01 ($\pm$0.00) | 0.01 ($\pm$0.00) | 0.01 ($\pm$0.00) | 0.01 ($\pm$0.00) | 0.01 ($\pm$0.00) | 0.01 ($\pm$0.00) |
| HardSATGEN (Li et al., 2023a) | N/A | N/A | N/A | N/A | **1915.25** ($\pm$**151.95**) | **47.83** ($\pm$**2.16**) | **137.69** ($\pm$**32.34**) | **1964.77** ($\pm$**98.36**) |
| MixSATGEN ($\lambda = 0$) | **103.62** ($\pm$**73.43**) | 443.61 ($\pm$594.34) | 1741.31 ($\pm$2306.26) | 35.07 ($\pm$72.85) | 0.91 ($\pm$1.41) | 0.01 ($\pm$0.00) | 0.01 ($\pm$0.00) | 0.01 ($\pm$0.00) |
| MixSATGEN ($\lambda = 0.1, \tau = 1$) | 61.18 ($\pm$67.41) | 485.24 ($\pm$542.34) | 1669.61 ($\pm$2355.00) | 54.45 ($\pm$86.88) | 13.22 ($\pm$15.02) | 0.01 ($\pm$0.00) | 0.01 ($\pm$0.00) | 0.01 ($\pm$0.00) |
| MixSATGEN ($\lambda = 0.5, \tau = 1$) | 684.30 ($\pm$944.48) | **970.01** ($\pm$**1356.16**) | **1355.34** ($\pm$**1851.32**) | 48.07 ($\pm$67.87) | 147.09 ($\pm$207.34) | 0.02 ($\pm$0.01) | 0.01 ($\pm$0.00) | 0.01 ($\pm$0.00) |
| MixSATGEN ($\lambda = 1.0, \tau = 1$) | 1675.33 ($\pm$2350.91) | 385.76 ($\pm$535.15) | 331.94 ($\pm$458.87) | **134.34** ($\pm$**189.10**) | 3.66 ($\pm$4.50) | 0.02 ($\pm$0.01) | 0.01 ($\pm$0.00) | 0.01 ($\pm$0.00) |
| MixSATGEN (random) | 1190.95 ($\pm$1912.65) | 851.71 ($\pm$1503.25) | 504.22 ($\pm$1002.57) | 100.23 ($\pm$248.36) | 0.52 ($\pm$0.75) | 0.01 ($\pm$0.00) | 0.01 ($\pm$0.00) | 0.01 ($\pm$0.00) |

Table 7: SBVA-Cadical runtime evaluation. STDs are in brackets. Best resemblance in **bold**.

| Method | hard-$a$ | hard-$b$ | hard-$c$ | hard-$d$ | hard-$e$ | hard-$f$ | hard-$g$ | hard-$h$ |
|---|---|---|---|---|---|---|---|---|
| Ground Truth | 3125.56 | 369.54 | 2630.29 | 59.89 | 4122.88 | 80.39 | 103.19 | 1079.08 |
| CA (Giráldez-Cru & Levy, 2015) | 0.12 ($\pm$0.01) | 0.13 ($\pm$0.00) | 0.06 ($\pm$0.00) | 0.05 ($\pm$0.00) | 0.06 ($\pm$0.00) | 0.12 ($\pm$0.04) | 0.05 ($\pm$0.00) | 0.06 ($\pm$0.00) |
| PS (Giráldez-Cru & Levy, 2017) | 0.81 ($\pm$0.02) | 0.80 ($\pm$0.02) | 0.07 ($\pm$0.00) | 0.07 ($\pm$0.00) | 0.07 ($\pm$0.00) | 0.07 ($\pm$0.00) | 0.06 ($\pm$0.00) | 0.06 ($\pm$0.00) |
| G2SAT (You et al., 2019) | 0.13 ($\pm$0.00) | 0.13 ($\pm$0.00) | 0.07 ($\pm$0.00) | 0.07 ($\pm$0.00) | 0.06 ($\pm$0.00) | 0.07 ($\pm$0.00) | 0.05 ($\pm$0.00) | 0.06 ($\pm$0.00) |
| GCN2S (Garzón et al., 2022) | 7324.38 ($\pm$10333.40) | 14.80 ($\pm$15.51) | 0.06 ($\pm$0.00) | 0.06 ($\pm$0.00) | 0.05 ($\pm$0.00) | 0.06 ($\pm$0.00) | 0.05 ($\pm$0.00) | 0.06 ($\pm$0.00) |
| HardSATGEN (Li et al., 2023a) | N/A | N/A | N/A | N/A | **4249.16** ($\pm$**303.18**) | **67.53** ($\pm$**7.34**) | **116.32** ($\pm$**7.37**) | **1254.20** ($\pm$**32.18**) |
| MixSATGEN ($\lambda = 0$) | **1048.88** ($\pm$**1396.85**) | 185.46 ($\pm$205.27) | **2113.74** ($\pm$**2638.67**) | 91.30 ($\pm$128.31) | 7.87 ($\pm$7.48) | 0.06 ($\pm$0.00) | 0.05 ($\pm$0.01) | 0.05 ($\pm$0.01) |
| MixSATGEN ($\lambda = 0.1, \tau = 1$) | 989.84 ($\pm$1371.68) | 219.44 ($\pm$186.88) | 6669.65 ($\pm$9425.98) | 87.56 ($\pm$127.11) | 40.14 ($\pm$44.90) | 0.12 ($\pm$0.04) | 0.10 ($\pm$0.03) | 0.11 ($\pm$0.03) |
| MixSATGEN ($\lambda = 0.5, \tau = 1$) | 303.93 ($\pm$400.04) | 11.37 ($\pm$14.30) | 959.93 ($\pm$1099.89) | **77.23** ($\pm$**93.80**) | 232.24 ($\pm$323.01) | 0.17 ($\pm$0.23) | 0.06 ($\pm$0.00) | 0.06 ($\pm$0.00) |
| MixSATGEN ($\lambda = 1.0, \tau = 1$) | 237.04 ($\pm$307.17) | **372.09** ($\pm$**272.73**) | 433.07 ($\pm$593.11) | 170.34 ($\pm$232.17) | 17.79 ($\pm$22.36) | 0.17 ($\pm$0.08) | 0.12 ($\pm$0.04) | 0.12 ($\pm$0.04) |
| MixSATGEN (random) | 986.48 ($\pm$1463.39) | 806.81 ($\pm$1145.72) | 825.09 ($\pm$1332.83) | 558.82 ($\pm$1282.94) | 3.58 ($\pm$1833.30) | 0.10 ($\pm$0.02) | 0.11 ($\pm$0.03) | 0.09 ($\pm$0.01) |

### A.2  PIPELINE EFFICIENCY

To demonstrate the efficiency advantage of MixSATGEN over previous state-of-the-art learning-based methods that generate from scratch, we compare G2SAT (You et al., 2019) with MixSATGEN

---

[2]The overall main-track winner of the 2023 SAT competition.

($\lambda = 0.1, \tau = 1.0$) for single instance generation, which is adopted as the standard training scheme in Garzón et al. (2022). As shown in Table 8, the pipelines consist of two processes: training and generating. The training process only needs to be performed once for a particular dataset, while the generating process can produce formulas repeatedly according to the target quantity. We measure the time for training and generating one specific formula with reference instances from the Hard dataset. The results show that MixSATGEN is significantly more efficient than G2SAT in both processes. Specifically, for the generating phase, MixSAT costs only approximately 1/2000 runtime compared to G2SAT.

Table 8: Consumed time of training and generating process for single instance generation.

| | Training | Generating |
|---|---|---|
| G2SAT (You et al., 2019) | 4467.05± 2315.81 | 2242.07±1280.28 |
| MixSATGEN | 1760.03±457.89 | 1.44±0.22 |

### A.3 SECONDARY INTERPOLATION

To evaluate the effect of secondary interpolation on the performance of formulas generated by MixSATGEN, we select two representative instance pairs and apply MixSATGEN($\lambda = 0.1, \tau = 1$) to obtain their $5\%$ mixing ratio formulas. We then use these formulas as the start formulas and interpolate them with their corresponding goal formulas, resulting in $10\%$ mixing ratio formulas. We compare these formulas with the original formulas, the $5\%$ start formulas, and the $10\%$ formulas obtained by one interpolation. The comparison is shown in Table 9, where we can see that the secondary interpolation does not degrade the performance of the formulas, as they maintain the same level of hardness as the one-interpolated formulas. This demonstrates the robustness of MixSATGEN in generating hard formulas through interpolation.

Table 9: Solver runtime comparison of formulas in secondary interpolation by MixSATGEN.

| | Ground Truth | 5% | Interpolated 10% | 10% |
|---|---|---|---|---|
| hard $b$-$a$ | 1382.71 | $1023.25 \pm 4.69$ | $1001.97 \pm 7.28$ | $1042.53 \pm 4.78$ |
| hard $d$-$c$ | 852.85 | $199.63 \pm 190.70$ | $6.13 \pm 3.14$ | $3.56 \pm 1.69$ |

### A.4 SATISFIABILITY PHASE PRESERVATION

The results of satisfiability phase preservation for generated formulas are presented in Table 10. It is observed that despite being trained on a dataset with a one-to-one ratio of satisfiable to unsatisfiable instances, previous methods exhibit a tendency to generate a specific satisfiability phase. Notably, instances generated by G2SAT, GCN2S, and HardSATGEN are all unsatisfiable, while PS generates exclusively satisfiable instances. Among all the methods, MixSATGEN maintains the closest phase ratio to the original reference instances.

Table 10: Satisfiability ratio of generated formulas and the preserve accuracy.

| Method | EASY | | | HARD | | |
|---|---|---|---|---|---|---|
| | UNSAT ratio | SAT accuracy | UNSAT accuracy | UNSAT ratio | SAT accuracy | UNSAT accuracy |
| Ground Truth | 50.00% | N/A | N/A | 50.00% | N/A | N/A |
| CA (Giráldez-Cru & Levy, 2015) | 87.5% | 0.00% | 75.00% | 75.00% | 0.00% | 50.00% |
| PS (Giráldez-Cru & Levy, 2017) | 0.00% | 100.00% | 0.00% | 0.00% | 100.00% | 0.00% |
| G2SAT (You et al., 2019) | 100.00% | 0.00% | 100.00% | 100.00% | 0.00% | 100.00% |
| GCN2S (Garzón et al., 2022) | 100.00% | 0.00% | 100.00% | 100.00% | 0.00% | 100.00% |
| HardSATGEN (Li et al., 2023a) | 100% | N/A | 100.00% | 100.00% | N/A | 100.00% |
| MixSATGEN | 50.00% | 83.33% | 83.33% | 43.41% | 38.19% | 25.00% |

## B PSEUDOCODE FOR MIXSATGEN

The algorithms for the training and generating process of MixSATGEN in Sec. 3 are presented in Alg. 1 and Alg. 2.

## C DATASET CONSTRUCTION

When selecting instances to construct the dataset for evaluation, we consider the following factors:

---

**Algorithm 1** Training process of MixSATGEN

---

**Input:** LCG pair $G_1, G_2$ with $2n_1, 2n_2$ literals; pretrained embedding model $D$; training epoch $t$;
**Output:** Optimized embedding model $D$;

   **for** $i = 1$ to $t$ **do**
       $\mathbf{H}_1 \leftarrow D(G_1)$
       $\mathbf{H}_2 \leftarrow D(G_2)$
       $\{(l_i, \neg l_i)\}^{n_1} \leftarrow \mathbf{H}_1, i \in n_1$
       $\{(l_j, \neg l_j)\}^{n_2} \leftarrow \mathbf{H}_2, j \in n_2$
       *// $\mathbf{M}_l$ is the InnerProduct*
       Calculate $\mathbf{M}_v$ where $\mathbf{M}_v[i,j] = \max\left(\mathbf{M}_l[l_i, l_j] + \mathbf{M}_l[\neg l_i, \neg l_j], \mathbf{M}_l[l_i, \neg l_j] + \mathbf{M}_l[\neg l_i, l_j]\right)$
       $\mathbf{S} \leftarrow \text{Sinkhorn}(\mathbf{M}_v)$
       $\mathbf{K} \leftarrow \text{AffinityMatrix}(G_1, G_2)$
       $loss \leftarrow -\text{vec}(\mathbf{S})^\top \mathbf{K} \text{vec}(\mathbf{S})$
       Update $D$ with $loss$
   **end for**

---

---

**Algorithm 2** Generating process of MixSATGEN

---

**Input:** LCG pair $G_1, G_2$ with $2n_1, 2n_2$ literals; optimized embedding model $D$; mixing ratio $\beta$; noise intensity factor $\lambda$; gumbel temperature $\tau$; **Output:** Interpolated formula $G_3$;

   *// Matching.*
   $\mathbf{H}_1 \leftarrow D(G_1)$
   $\mathbf{H}_2 \leftarrow D(G_2)$
   $\{(l_i, \neg l_i)\}^{n_1} \leftarrow \mathbf{H}_1, i \in n_1$
   $\{(l_j, \neg l_j)\}^{n_2} \leftarrow \mathbf{H}_2, j \in n_2$
   *// $\mathbf{M}_l$ is the InnerProduct*
   Calculate $\mathbf{M}_v$ where $\mathbf{M}_v[i,j] = \max\left(\mathbf{M}_l[l_i, l_j] + \mathbf{M}_l[\neg l_i, \neg l_j], \mathbf{M}_l[l_i, \neg l_j] + \mathbf{M}_l[\neg l_i, l_j]\right)$
   $\mathbf{\Gamma} \sim \text{Gumbel}$
   $\mathbf{S} \leftarrow \text{Sinkhorn}\left((\mathbf{M}_v + \lambda\mathbf{\Gamma})/\tau\right)$    *// Gumbel-Sinkhorn*
   $\mathbf{P} \leftarrow \text{Hungarian}(\mathbf{S})$
   Extract correspondence map and confidence score of variables from $\mathbf{P}$
   *// Mixing.*
   $G_3 \leftarrow G_1$
   *// Global confidence for every clause in $G_1$.*
   Calculate the global confidence score of every clause $c_1$ in $G_1$ with sum of the confidences of its variables
   **while** the substituted clause ratio $\leq \beta$ **do**
       Pick $c_1$ in descending order of global confidence score over $G_1$
       *// Local confidence for related clause in $G_2$.*
       Trace to clause set $C_2$ on $G_2$ whose item has shared variables with $c_1$ through correspondence map
       Calculate the local confidence score of clause $c_2$ in $C_2$ with sum of the confidences of its shared variables with $c_1$ through correspondence map
       $C_2 \leftarrow$ clauses with the highest local confidence score in $C_2$
       *// Global confidence for remained clause in $G_2$.*
       Calculate the global confidence score of $c_2$ in $C_2$ with sum of the confidences of its variables
       $C_2 \leftarrow$ clauses with the highest global confidence in $C_2$
       $c_2 \leftarrow$ take a clause from $C_2$ randomly
       *// Clause replacement for graph mixup.*
       Remove all edges of $c_1$ on $G_3$
       $L \leftarrow \{\text{literal } l \in G_3\}$ where the correspondence literal of $l$ links to $c_2$ on $G_2$
       Add edges between $l \in L$ and $c_1$ on $G_3$
   **end while**

---

1) It should include instances derived from real-world application scenarios to ensure the practical value of our experimental validation. Therefore, we choose instances from SAT Competition benchmarks, which are encoded from real-world scenarios.

2) We aim to maintain an equal proportion of satisfiable and unsatisfiable instances in our data to verify the general applicability of MixSAT. Hence, we maintain a 1:1 ratio of sat to unsat instances in the dataset.

3) Recognizing the significance of computational difficulty in experimental validation, we specifically select instances with higher computational hardness to constitute the Hard dataset.

4) We select formulas that have a suitable scale range of around hundreds of variables and hundreds to thousands of clauses. While current graph network based methods may have difficulty scaling to very large graphs, MixSATGEN also faces the same scale constraint due to the scale-related resource-consuming affinity matrix construction in training process.

