# OpenReview forum: "MixSATGEN: Learning Graph Mixing for SAT Instance Generation"
_ICLR.cc/2024/Conference — ICLR 2024 poster_

### Official Review · Reviewer_SBgK · 2023-10-30

**Soundness:** 2 fair
**Presentation:** 3 good
**Contribution:** 2 fair
**Rating:** 5
**Confidence:** 3

**Summary:**

This paper proposes a method for automatically generating SAT instances. Automatic generation of SAT instances is highly demanded since we need a large amount of SAT instances for estimating parameters of learning-based SAT solvers or tuning hyperparameters of traditional SAT solvers. While previous state-of-the-art learning-based SAT instance generation methods generate instances from a single reference, the proposed algorithm, MixSAT, generates SAT instances by interpolating two SAT instances. MixSAT first makes an assignment map of variables of two SAT instances. Then, it uses the map to select similar clauses and finally exchanges these similar clauses to make new SAT instances. On making an assignment map, the proposed method puts random noise for preserving difficulties. Experimental results show that MixSAT can preserve more graph structural properties than baseline methods, except for UNSAT instances. Moreover, MaxSAT preserves the difficulty of SAT instances, in hard satisfiable instances.

**Strengths:**

**Important topic:**
I agree with the motivation that we need more real-world SAT instances to improve the performance of Boolean SAT solvers. I agree that to generate SAT instances, preserving difficulty is a challenging task since SAT instances seem sensitive to small changes.

**Clearly written paper:**
The paper is generally clearly written and easy to read. Figures 1-3 help us understand the complex procedure of MaxSAT.

**The idea of using interpolation is interesting:**
The idea of using interpolation is interesting. It seems a reasonable way of data augmentation. Moreover, the proposed method seems
carefully designed to perform interpolation between two CNFs.

**Experimental results show the superiority of the proposed method:**
Although not always better than HardSATGEN, the proposed method outperforms other baseline methods.

**Weaknesses:**

**Experimental results are not strong:**
1. Experimental results show that the proposed method is not always superior to HardSATGEN. Especially, the running time evaluation results in Tab.2 show that the proposed method fails to preserve the computational difficulties of hard instances. The results seem to contradict the claim of the paper.
1. Li et al. (2023) reported that hyperparameter tuning with instances generated by HardSATGEN improves performance on real-world problems. Since a primal objective of generating SAT instances is to tune the parameters of solvers, the paper should report the results of using the instances generated by MixSAT to tune the parameters of existing solvers.
1. Runtime evaluations were conducted with CaDiCaL. I think the paper should also report the performance of different solvers like Kissat since different solvers sometimes show completely different performances. In performance evaluation, Li et al. (2023) compared the performance of three different solvers. It seems a safer way to evaluate the running time of SAT solvers.



**Some important details of experiments are missing:**
Some important details of experiments are not reported in the paper:
1. The proposed method generates instances by interpolating two instances. Therefore, the performance of the proposed method depends on how we make a pair. However, the details of making pairs are not reported in the paper.
2. It is said that the dataset was taken from SATLIB and SAT Competition 2021. However, how the authors select easy and hard [a-h] instances from a large amount of instances seems not explained in the paper. The paper should show the reason behind selecting these instances to make the evaluation fair.



**There are claims not supported by experimental results:**
Some claims of the paper seem not supported by the experimental results:
1. The paper says the proposed method is more efficient (page 2), but no experimental results support the claim.
2. The paper says that MaxSAT introduces some randomness to maintain computational hardness. However, experimental results reported in Table 2 show that randomness does not help much in preserving hardness.

These points make me feel that the proposed approach is not well-motivated.

**Questions:**

I'd be happy if the authors addressed my concerns mentioned in the weakness section.

---

> ### Author Response · Authors · 2023-11-19
> **Response by Authors (1/3)**
>
> Thanks for the thorough review and valuable comments. We are encouraged with your acknowledgment of our topic importance, writing clarity, methodology design, and empirical results. Below we respond to your specific comments.
>
> > **Q1: Experimental results show that the proposed method is not always superior to HardSATGEN.**
>
> Please note that reproducing computational hardness in the SAT instance generation task is considerably challenging which has not been achieved until HardSATGEN [1]. Its success stems from its fine-grained control over the instance structure, particularly for the unsatisfiable core structures, as well as the postprocessing design involved to maintain the dominant effect of the unsat core structure.
>
>
> While the design in [1] is effective, it confines the method to unsatisfiable instances only. Since real-world datasets generally incorporate both satisfiable and unsatisfiable instances, merely learning from the unsat instances can lead to certain negative impacts on downstream tasks like hyperparameter tuning, as evidenced by Table 3 in the new revision. In contrast, MixSAT surpasses this limitation, demonstrating generality to both satisfiable and unsatisfiable instances. MixSAT is also free from the node merge-split framework of previous learning-based methods [1-3], naturally meeting the hardness analysis and requirements set forth in [1].  Notably, our performance on satisfiable instances is significantly superior to all other methods.
>
> We acknowledge that our effectiveness on unsatisfiable instances is not as strong, especially considering [1] already provides a tailored solution for unsatisfiable problems. However, we believe the two methods can complement each other. As evidence, we have supplemented the study with hyperparameter optimization experiments and present the combined effects of both methods in Table 3, which show the significant effectiveness of our methods in practice as a boosting tool complementary to HardSATGEN.
>
>
> > **Q2: Hyperparameter tuning with the generated instances.**
>
> Thanks for the valuable point. We supplement the hyperparameter tuning results on the state-of-the-art SAT solver Kissat in Table 3 of the new manuscript. Due to the complexity of the table format, we refrain from presenting it in this response.
>
> We adopt one of the Bayesian optimizers HEBO to tune the solver. The tuned hyperparameters include restart interval, reduce interval, and decay, which pose impacts on the frequency of restarting, frequency of learned clause reduction, and per mille scores decay respectively. For each method, the Bayesian optimizer tries 200 sets of hyperparameters and we select the top 5 hyperparameters to evaluate on test instances. The best runtime results are presented in the table. Due to the costly Bayesian optimization process, we select instances with runtimes under 500 seconds for experiments.
>
> Note in Table 2, MixSAT reproduces computational hardness well on satisfiable instances, while HardSATGEN [1] can only learn from unsatisfiable instances. Table 3 shows the matched performance where MixSAT and HardSATGEN perform the best for satisfiable and unsatisfiable instances respectively. Thus, we also consider simultaneously adopting MixSAT learned from satisfiable instances and HardSATGEN learned from unsatisfiable instances for tuning, which presents the best overall performance. In general, the generated instances by the proposed methods bring a significant solver runtime reduction of over 50% to the solver.
>
>
> > **Q3: Runtime evaluations conducted with more modern solvers.**
>
> Thanks for the comment. We supplement runtime evaluation on Kissat SBVA-Cadical (the overall main-track winner of the 2023 SAT competition) solvers over the generated formulas as well as the original benchmarks in Tables 6 and 7 of the new revision. We set the runtime limit for Kissat at 5000 seconds, while for SBVA-Cadical, no runtime limit is imposed due to its slower execution on the Hard dataset. As seen, MixSAT performs consistently regarding the computational hardness across different solvers.

---

> ### Author Response · Authors · 2023-11-19
> **Response by Authors (2/3)**
>
> > **Q4: The performance of the proposed method depends on how we make a pair. However, the details of making pairs are not reported in the paper.**
>
>
> Firstly, it is essential to note that the core methodology of our approach centers around the design of the interpolation method, which is non-trivial when applied to graph structures, especially in the case of large-scale bipartite graphs with distinctive structures. In the paper, we elaborate on how we design the graph interpolation method to ensure consistency in generating data that aligns with the original distribution, in structural (through matching) and hardness (through matching with randomness) aspects.
>
>
> On the other hand, of course pair selection method is an integral part of our methodology. The process of how we filter pairs has been detailed in Sec. 2.3 of the original paper, with the specific design outlined as follows:
>
> We introduce an entropy-based filter to first select potentially valuable pairs for mixing (since the similarity of different pairs can vary much, not all pairs can yield quality matching results), which assesses and filters pairs using the row-wise entropy of the soft alignment matrix $\mathbf{S}\in\mathbb{R}^{n_1\times n_2}$:
> $$h = -\sum_{i=0}^{n_1} \sum_{j=0}^{n_2}s'\_{ij}\log{s'\_{ij}} \quad \text{where} \quad s'\_{ij} = \frac{s\_{ij}}{\sum\_{k}s\_{ik}}$$
> which implies the sharpness of the soft alignment matrix between $G_1$ and $G_2$. A smaller value of $h$ denotes more distinct matching relations, leading to a potentially better pair for mixing.
>
> > **Q5: The reason behind selecting these instances as datasets to make the evaluation fair.**
>
> When selecting instances to construct the dataset for evaluation, we consider the following factors:
>
> 1) It should include instances derived from real-world application scenarios to ensure the practical value of our experimental validation. Therefore, we choose instances from SAT competition benchmarks, which are encoded from real-world scenarios.
>
> 2) We aim to maintain an equal proportion of satisfiable and unsatisfiable instances in our data to verify the general applicability of MixSAT. Hence, we maintain a 1:1 ratio of sat to unsat instances in the dataset.
>
> 3) Recognizing the significance of computational hardness in experimental validation, we specifically select instances with higher computational hardness to constitute the Hard dataset.
>
>
> > **Q6: Experimental support for "the proposed method is more efficient".**
>
>
> To demonstrate the efficiency advantage of MixSAT over previous state-of-the-art learning-based methods that generate from scratch, we compared G2SAT[2] with MixSAT ($\lambda=0.1, \tau=1.0$) for single instance generation, which is adopted as the standard training scheme in [3].
>
>
> The pipelines consist of two processes: training and generating. The training process only needs to be performed once for a particular dataset, while the generating process can produce formulas repeatedly according to the target quantity. We measure the time for training and generating one specific formula. As shown below and also in Table 8 of the updated revision,  MixSAT is significantly more efficient than G2SAT in both processes. Specifically, for the generating phase, MixSAT costs only approximately 1/2000 runtime compared to G2SAT.
>
>
> |       |     **Training**      |   **Generating**    |
> |:----------:|:---------------------:|:-------------------:|
> | **G2SAT**  | 4467.05 $\pm$ 2315.81 | 2242.07 $\pm$ 1280.28 |
> | **MixSAT** | 1760.03 $\pm$ 457.89  |   1.44 $\pm$ 0.22   |

---

> ### Author Response · Authors · 2023-11-19
> **Response by Authors (3/3)**
>
> > **Q7: Experimental support for "MixSAT introduces some randomness to maintain computational hardness".**
>
>
> Note Table 2 illustrates the average solver runtime of all pairs across each instance. In our analysis, a tunable level of randomness is introduced to prevent hardness degradation caused by the interaction and conflicts of substructures. However, some deterministically generated formulas ($\lambda=0$) have already effectively maintained the hardness, surpassing the entirely random ones (due to inherent randomness). Consequently, the average solver runtime cannot effectively reflect the improvement of hardness led by introducing randomness.
>
> To straightforwardly reiterate the effect of randomness on the computational hardness of generating formulas, below and also in Table 5 of the new revision presents the results on instance pairs where MixSAT exhibits some degree of hardness degradation. We gradually increased the $\lambda$ value to enhance the randomness. As seen, when the randomness increases,  the hardness of the initially degraded pairs is recovered to a certain extent compared with the ground truth. Note although pair $e$-$f$ fails to maintain hardness on both deterministic ($\lambda=0$) and purely random generations, it gains some hardness when partial randomness is introduced ($\lambda=0.5$).
>
> | **Randomness**      | **pair $a$-$c$**  | **pair $b$-$d$**    | **pair $d$-$c$**   | **pair $e$-$f$** |
> |:-------------------:|:-----------------:|:-------------------:|:------------------:|:----------------:|
> | **Ground Truth**    | 254.22            | 1382.71             | 852.85             | 2629.03          |
> | **$\lambda=0$**     | 0.09 $\pm$ 0.01     | 52.73 $\pm$ 3.87      | 117.83 $\pm$ 129.39  | 0.31 $\pm$ 0.01    |
> | **$\lambda=0.1$**   | 0.29 $\pm$ 0.01     | 138.11 $\pm$ 18.68    | 199.63 $\pm$ 190.70  | 33.95 $\pm$ 0.03   |
> | **$\lambda=0.5$**   | 0.19 $\pm$ 0.02     | 23.14 $\pm$ 0.74      | 122.32 $\pm$ 12.33   | 500.12 $\pm$ 4.60  |
> | **$\lambda=1.0$**   | 3.59 $\pm$ 0.06     | 442.11 $\pm$ 3.53     | 188.93 $\pm$ 18.49   | 21.42 $\pm$ 2.41   |
> | **$\lambda=10.0$**  | 39.88 $\pm$ 6.91    | 332.05 $\pm$ 25.57    | 1530.78 $\pm$ 838.56 | 2.05 $\pm$ 1.80    |
> | **$\lambda=100.0$** | 130.32 $\pm$ 63.88  | 472.43 $\pm$ 13.31    | 1429.93 $\pm$ 308.41 | 0.69 $\pm$ 0.68    |
> | **Random**          | 215.73 $\pm$ 120.95 | 1116.42 $\pm$ 1203.59 | 695.24 $\pm$  799.58  | 11.49 $\pm$ 9.12   |
>
>
> ---
>
> We hope this response could help address your concerns. We sincerely wish that you could reconsider the novel contribution and potential impact of our work on the community. We look forward to receiving your valuable feedback soon.
>
> ---
>
> **Reference:**
>
> [1] HardSATGEN: Understanding the Difficulty of Hard SAT Formula Generation and A Strong Structure-Hardness-Aware Baseline. KDD 2023.
>
> [2] G2SAT: Learning to Generate SAT Formulas. NeurIPS 2019.
>
> [3] On the Performance of Deep Generative Models of Realistic SAT Instances. SAT 2022.

---

> > ### Comment · Reviewer_SBgK · 2023-11-21
> > **Thank you for the response. I have some additional questions.**
> >
> > Thank you for the very detailed response. I have some additional questions:
> >
> > 1. The new results in Table 3 are interesting. Could you please tell me more details about the experimental settings? How do you prepare training and test data?
> >
> > 2. Table 8 shows a significant improvement over G2SAT.  Why do you compare with G2SAT? What happens if we compare MixSAT with other baselines?

---

> ### Author Response · Authors · 2023-11-21
> **Response to Reviewer SBgK**
>
> We sincerely appreciate your prompt reply and the acknowledgment of our response. Below we respond to your new questions.
>
> > Q1: The new results in Table 3 are interesting. Could you please tell me more details about the experimental settings? How do you prepare training and test data?
>
> We adhere closely to HardSATGEN's experimental settings of its solver tuning experiment. The data used for tuning the solver aligns with the datasets employed in previous experiments concerning graph structure (Sec. 3.2) and computational hardness (Sec. 3.3). The training set is consistent with previous experiments, specifically the HARD dataset. Trained models are then applied to generate new instances based on reference instances hard-a, hard-d, hard-f, and hard-g,  which are selected from the HARD dataset with runtimes of less than 500 seconds. This selection is motivated by the very time-intensive nature of hyperparameter optimization, involving testing all augmented instances with 200 sets of hyperparameters. Opting for instances with shorter runtimes, such as the aforementioned four, helps mitigate experiment overhead. The reference instances collectively form the test data. The underlying expectation is that the augmented data could effectively guide the solver to enhance its performance on the original data.
>
> > Q2: Table 8 shows a significant improvement over G2SAT. Why do you compare with G2SAT? What happens if we compare MixSAT with other baselines?
>
>
> Previous learning-based methods, such as G2SAT, GCN2S, and HardSATGEN, share a common foundation built on the node merge-split framework brought out by G2SAT, employing the same main generation process involving a sequence of node-merging operations (differing mainly in the use of differently trained networks to select the node pairs for merging). Consequently, the runtime results of G2SAT serve as the representative for these approaches operating within the same paradigm. The remaining baselines under the split-merge paradigm show no significant difference in terms of time efficiency compared to G2SAT.
>
> Given that the efficiency comparison primarily centers around learning-based methods, traditional hand-crafted methods are not included in Table 8. However, we can still provide discussions of these methods in this response below:
>
> Traditional generative methods including CA and PS can generate instances within mere seconds and require no training. They are more efficient than MixSAT. However, since the paradigm of these methods basically follows the pipeline of first identifying one or two structural metrics and then specifying an algorithm capable of controlling these metrics, these models fail to unravel specific data characteristics adaptively, merely matching partial structural metrics through manual control, as discussed in G2SAT and HardSATGEN. This can also be evidenced by Tables 1-3.
>
> ---
>
> We hope this response could help address your concerns, and look forward to receiving your feedback soon.

---

> > ### Comment · Reviewer_SBgK · 2023-11-22
> > **Thank you for answering my questions.**
> >
> > Thank you for the response. I appreciate the hard work of the authors, and I feel that the paper is significantly improved. However, there remain some concerns:
> >
> > 1. The experimental results on runtime evaluation (Table 2) are not strong. This result mismatches with the claim described in the introduction. In the introduction, the proposed method is said to achieve hardness reproducing and structure resemblance. However, the MixSAT does not work well on preserving the hardness of UNSAT instances, and it is unclear why it does not work well.
> >
> > 2. The proposed method is evaluated with a few instances whose selection procedure (Appendix D) seems somewhat subjective. From the information described in Appendix D, I cannot judge whether the selection process of instances is fair or not.
> >
> > 3. The newly added results on parameter tuning are interesting. However, I think the experimental settings explained by the authors are not convincing since they use instances augmented from hard-[adfg] for tuning parameters and evaluate the performance of the tuned models using the original hard-[adfg] as the test data. This is unrealistic since we usually cannot access test data when we train a model; it seems easy to select the best hyperparameters if we know the test instances when we tune the parameter. The paper should use different instances for training and testing.

---

> ### Author Response · Authors · 2023-11-22
> **Further Response to Reviewer SBgK**
>
> We sincerely appreciate your valuable engagement and the time you dedicated to reviewing our work. However, we may hold differing opinions on certain points you raised. Below we respond to your concerns.
>
> > **The experimental results on runtime evaluation (Table 2) are not strong. This result mismatches with the claim described in the introduction. In the introduction, the proposed method is said to achieve hardness reproducing and structure resemblance.**
>
> We admit that Table 2's results are not strong. But as we noted before, the resemblance in computational properties is very challenging (as listed as one of the ten key challenges in propositional reasoning and search [1]), and in fact MixSAT's results have already been the best among all other hand-crafted methods and learning-based methods **with general applicability**. Please note again HardSATGEN is a tailored method for UNSAT instances, whose success is at the cost of its ability to learn from SAT instances. Moreover, we respectfully believe underperforming one baseline under a specific task for partial data should not directly lead to rejection. We have revised the expression "achieve hardness reproducing" in the introduction to "achieve better hardness resemblance" to more accurately align with the observed empirical results.
>
> [1] Ten challenges in propositional reasoning and search. IJCAI 1997.
>
> > **The proposed method is evaluated with a few instances whose selection procedure (Appendix D) seems somewhat subjective. From the information described in Appendix D, I cannot judge whether the selection process of instances is fair or not.**
>
>
> We assert that the principles outlined in Appendix D are devoid of any subjective bias. We never cherry-picked the instances with the aim of maximizing MixSAT's performance. We are in the process of organizing our code and plan to upload it before the rebuttal deadline to ease your concerns. The rationale behind the three rules of selection in Appendix D is as follows:
>
> 1) The objective is to augment real-world SAT data to solve the data bottleneck in real-world scenarios, it is natural to select real-world data rather than random instances.
>
> 2) To test the applicability generality of the methods (i.e. performance on SAT and UNSAT sources), we select an equal number of SAT and UNSAT instances to compose the datasets. This is natural to approximate real-world application data that encompasses both SAT and UNSAT instances (otherwise we may not need a SAT solver to decide instances' satisfiability).
>
> 3) The imposition of a hardness requirement serves to guarantee the verifiability of the solver runtime experiment. We initially selected the dataset of G2SAT to evaluate our methods, but we observed that most of its instances could be solved in nearly 0 seconds, which made the solver runtime comparison meaningless. Thus, for solver runtime evaluation, we select hard instances from G2SAT and supplement additional instances from SAT competition to form the Hard dataset. While the instances in the Easy dataset are all from G2SAT.
>
>
> We consider these selections to be fundamental and intuitive requirements. If you are still confused about some of the specific selection rules, please let us know.
>
> > **However, I think the experimental settings explained by the authors are not convincing since they use instances augmented from hard-[adfg] for tuning parameters and evaluate the performance of the tuned models using the original hard-[adfg] as the test data.**
>
> This design is out of the consideration that the objective is to augment SAT instances that follow the original data distribution, thereby solving the data bottleneck. Please kindly note that this is not a training-testing paradigm, but rather a reference-augmentation paradigm similar to Tables 1 and 2 for structure and hardness resemblance evaluation.  The performance of the solver tuned on instances augmented from hard-[adfg] and tested on the original hard-[adfg] instances shows the higher-level resemblance of the generated instances to the reference instances (obeying the same underlying distribution), such that the augmented data could effectively guide the solver to enhance its performance on the original data sources. This achievement is non-trivial, as can be evidenced by the failures of all previous methods in this setting including HardSATGEN. Please also note that this experimental setting exactly follows the solver tuning results in HardSATGEN.
>
> ---
>
> We hope this response could help address your remaining concerns. We believe our work could contribute to the community as a novel framework for SAT instance generation, distinct from previous methods adhering to the split-merge framework. We sincerely wish that you could reconsider the methodology novelty, efficiency superiority, and general empirical superiority on structure and hardness, as well as the potential impact of our work on the community. We look forward to receiving your feedback soon.

---

> > ### Comment · Reviewer_SBgK · 2023-11-22
> > **Thank you for the further response**
> >
> > Thank you for the further response.
> >
> > ---
> >
> > The data selection procedure explained by the authors is the following:
> >
> > 1.  The number of SAT/UNSAT instances is equal.
> > 2.  Select real-world and hard instances.
> >
> > Since these rules would not be enough to identify the hard instances used in experiments from SATLIB and SAT 2021,  I think that there are some implicit rules in the selection procedure.
> >
> > ---
> >
> > I also agree that the results of the hyperparameter tuning task are non-trivial. However, these results do not show that the proposed method can be used to solve the data scarcity problem described in the introduction:
> > > Both model training for modern learning-based SAT solvers or hyperparameter tuning for traditional solvers require relevant instances as many as possible to either tune the trainable model parameters or the hyperparameters. Unfortunately, SAT instances are often scarce in practice
> >
> > I think both use cases described above follow a training-testing paradigm.

---

> ### Author Response · Authors · 2023-11-23
> **Further Response to Reviewer SBgK**
>
> Thank you for your timely reply and your valuable engagement in this tight rebuttal window, which helps a lot for us to refine our paper.
>
> > **Since these rules would not be enough to identify the hard instances used in experiments from SATLIB and SAT 2021, I think that there are some implicit rules in the selection procedure.**
>
> Indeed, our instances are originally sourced from G2SAT, a dataset derived from SATLIB and SAT 2021. The Easy dataset is directly from G2SAT. The Easy dataset directly originates from G2SAT. However, given the scarcity of challenging instances in G2SAT (only 4 hard instances), we have supplemented the remaining 4 instances from SAT 2021, ensuring they share similar problem scales with G2SAT instances. The underlying implicit rule may be that current graph network based methods may not be able to scale to very large graphs (SAT 2021 often contains problems with 10k variables and 100k clauses). While also, our affinity matrix construction is somehow resource-consuming, thus MixSAT still cannot exceed the scale limit of current learning-based methods. Thanks again for noting, we are sorry that we omitted this underlying scale requirement, we will supplement this in Appendix D. On the other hand, we have uploaded our code to https://anonymous.4open.science/r/MixSAT/, which can be evaluated by any data to prove MixSAT's performance.
>
> > **I also agree that the results of the hyperparameter tuning task are non-trivial. However, these results do not show that the proposed method can be used to solve the data scarcity problem described in the introduction.**
>
> The ultimate objective of SAT instance generation is to generate instances that resemble the original data, regarding structure and hardness. As guided in *Ten challenges in propositional reasoning and search (IJCAI 1997)*:
>
> "CHALLENGE 10: Develop a generator for problem instances that have computational properties that are more similar to real-world instances."
>
> We believe the requirement of the resemblance between the generated and original data is sufficient to resolve the data scarcity problem. After all, that is the whole point of SAT instance generation methods in the first place. Our newly supplemented solver tuning results can support this data resemblance on a higher level, which surpasses all other baselines. From the application perspective, we admit it should better follow the training-testing paradigm, while our current solver tuning results may not be supportive enough for this point. In your initial review, you mentioned HardSATGEN as an example for the hyperparameter tuning experiments. It is per your request to follow its setting and supplement the new results to show the superiority of MixSAT. We believe the new results are still meaningful and can strengthen our paper as evidence for a higher level of resemblance between the generated and original data. Please also note that previous baselines are all ineffective for the current solver tuning setting, let alone the solver tuning performance for application purposes. We will continue to make efforts to another version of solver tuning results, but since the discussion window is closing, we wonder whether our efforts in the rebuttal period have met your requests and addressed most of your concerns, we would sincerely appreciate it if you could reconsider the current overall paper contribution and the massive new results supplemented in this very tight rebuttal window to strengthen our paper.
>
> Thanks again for your interest and dedicated efforts in reviewing our work. We truly appreciate that and look forward to receiving your feedback soon.

---

### Official Review · Reviewer_5Kyc · 2023-10-31

**Soundness:** 2 fair
**Presentation:** 3 good
**Contribution:** 2 fair
**Rating:** 5
**Confidence:** 3

**Summary:**

This paper is devoted to machine learning assisted generation of
families of benchmark formulas for propositional satisfiability (SAT)
solvers. The paper builds on the use of graph representations of
formulas in conjunctive normal form (CNF) and utilizes a graph
interpolation approach, which modifies the graph structure by
replacing some of its substructures from other similar instances. The
paper argues that the proposed approach is able to maintain the
original hardness of the corresponding families of benchmarks, thanks
to the use of Gumpel noise. The presented experimental results claim
to demonstrate the advantage over the state of the art in SAT
benchmark generation from the perspective of (1) maintaining the
structure of the benchmark family and (2) its hardness.

**Strengths:**

- The entire flow of the proposed approach are provided one after
  another.
- The overall presentation is nice, especially the figures look
  beautiful and help a reader understand the ideas.
- To the best of my understanding, the proposed approach is novel.

**Weaknesses:**

- Although all the steps of the approach are listed, they are not
  augmented with clear arguments for why they are used. This seems to
  be a standard issue with all the works on applying ML methods in/for
  combinatorial problem solvers where the authors propose to apply
  ~50-100 steps with no clear justification. This trend is clearly
  detached from the mainstream SAT research where each idea has to be
  clearly articulated and justified. Granted it may be just me getting
  lost in matters that are straightforward to any ML expert.

- Experimental results look rather weak to me as they seem to be a
  mixed bag. At least, I don't see a clear advantage of the proposed
  approach over the state of the art in terms of the hardness of the
  generated formulas.

- Minor: the authors say that SAT is a combinatorial optimization (CO)
  problem while it isn't. There is no optimization in the original
  decision formulation of SAT.

- Minor: I would suggest rewording the sentence (in the introduction):
  "resemble the computational complexity". Computational complexity of
  SAT is well understood.

**Questions:**

- A key question is this line of research is whether the formulas
  generated automatically exhibit the properties intrinsic to
  "industrial instances". This is important given the widespread use
  of modern SAT solvers in industrial settings. So I would like to ask
  the authors whether the ability of their approach to capture this.

- Can you comment on the rationale of (at least some of) the steps
  your approach performs?

---

> ### Author Response · Authors · 2023-11-19
> **Response by Authors (1/2)**
>
> Thanks for the valuable comments, and recognition of our presentation and novelty. Below we respond to your specific comments.
>
> > **Q1: The rationale of the steps the approach performs.**
>
> Thanks for the note. We totally agree with your assertion that the rationale behind every effort of the methodology design should be articulated and elucidated in the paper.
>
> We believe this principle should be a consensus across both the mainstream SAT research and the ML4SAT research. In fact, we try to make efforts to achieve such clarity in the original paper. Indeed, we acknowledge that certain points may not have been sufficiently clarified, leading to confusion. Here we would like to elucidate the underlying rationale for each design choice in our methodology. We hope this clarification addresses your concerns and fortifies the overall clarity of our work.
>
> - **Why mixing?** Previous node split-merge framework for SAT instance generation exhibits poor performance in computational hardness resemblance. Though more fine-grained controls in [1] on structures can improve the performance, it also introduces applicability limitations. Thus we resort to a different graph interpolation approach that directly manipulates the raw structures of SAT instances in place, which is achieved by mixing substructures of SAT instances. This design is more efficient, more adaptive, and can naturally avoid the hardness degradation causes in the split-merge framework as analyzed in [1]. (Page 2)
>
> - **Why matching?** The exchanged substructures should be matched such that the mixing process can largely preserve the original essential patterns and the mixed instances remain within the underlying distribution of reference instances regarding both structural similarity and computational hardness. (Page 3)
>
> - **Why adopt the neural SAT solving task as a pretraining task?** It is for the sake of hardness-awareness. This pretraining task enables the model to not only capture the graph structure but also naturally discover the inherent properties regarding the computational hardness of the instances. (Page 4)
>
> - **Why perform online optimization with matching loss?** The pretraining task guarantees plausible initial node features, but it is not directly related to the matching purpose. The online optimization with matching loss is to directly maximize the similarity between the node features of the two SAT instances. (Page 5)
>
> - **Why introduce randomness?** Learning-based methods excel at identifying similarities within structures. With the replacement on top of the learned variable correspondence map, similar substructures from two reference graphs can easily interact and probably lead to conflicts and consequent hardness degradation. Thus, beyond the regular matching pipeline, a tunable level of randomness could avoid hardness degradation. (Page 5)
>
> - **Why use Gumbel trick to introduce randomness?** Introducing randomness to assignments is non-trivial, while Gumbel trick on discrete assignment distributions is well studied in [4] with satisfying sampling property that $H(\mathbf{M}+\mathbf{\Gamma})\sim\mathcal{G.M.}(\mathbf{M})$, where $\mathbf{\Gamma}$ denotes a Gumbel noise matrix, $\mathcal{G.M.}(\mathbf{M})$ is the Gumbel-Matching distribution with parameter $\mathbf{M}$, and $H(\cdot)$ denotes the matching operator providing the best matching results, which can be achieved by the Hungarian algorithm for the linear matching case. (Page 5)
>
> - **Why this mixing algorithm?** The mixing algorithm we propose in this paper is simple and natural. It directly utilizes the variable matching confidence obtained in the matching process, which increases the probability of replacing similar structures. We perform substitutions at the clause level because clauses are the smallest substructure of the SAT formula, thus they correspond to the smallest interpolation step in the interpolation scenario. (Page 6)
>
> We have revised the manuscript accordingly to increase the presentation clarity of our paper. Please let us know if you have any other confusion regarding the approach rationale.

---

> > ### Comment · Reviewer_5Kyc · 2023-11-21
> > **Reply to the authors**
> >
> > Thanks for the reply and for revising the paper. I appreciate the effort you've put in this.

---

> ### Author Response · Authors · 2023-11-19
> **Response by Authors (2/2)**
>
> > **Q2: Experimental results on hardness seem weak.**
>
> Although MixSAT performs weaker than HardSATGEN [1] on unsat instances, it outperforms other methods. Please note that reproducing computational hardness in the SAT instance generation task is considerably challenging which has not been achieved until [1]. Its success stems from its fine-grained control over the instance structure, particularly for the unsatisfiable core structures, as well as the postprocessing design involved to maintain the dominant effect of the unsat core structure.
>
> While the design in [1] is effective, it confines the method to unsatisfiable instances only. Since real-world datasets generally incorporate both satisfiable and unsatisfiable instances, merely learning from the unsat instances can lead to certain negative impacts on downstream tasks like hyperparameter tuning, as evidenced by Table 3 in the current revision. In contrast, MixSAT surpasses this limitation, demonstrating generality to both satisfiable and unsatisfiable instances. MixSAT is also free from the node merge-split framework of previous learning-based methods [1-3], naturally meeting the hardness analysis and requirements set forth in [1].  Notably, our performance on satisfiable instances is significantly superior to all other methods.
>
> We acknowledge that our effectiveness on unsatisfiable instances is not as strong, especially considering [1] already provides a tailored solution for unsatisfiable problems. However, we believe the two methods can complement each other. As evidence, we have supplemented the study with hyperparameter optimization experiments and present the combined effects of both methods in Table 3, which shows the significant effectiveness of our methods in practice as a boosting tool complementary to HardSATGEN.
>
> > **Q3: Whether the formulas generated automatically exhibit the properties intrinsic to "industrial instances"?**
>
> The data we adopt in our experiments are exactly derived from real-world scenarios (from SAT Competition Benchmarks). Note the disparity between industrial data and real-world data can be viewed as minimal, as both originate from practical application scenarios. The experiments are designed to evaluate whether the generated instances exhibit the properties intrinsic to real-world instances. The evaluation for the real-world resemblance is two-fold: 1) structural properties compared to the reference real-world instances (Sec. 3.2); 2) computational properties compared to the reference real-world instances (Sec. 3.3). In the rebuttal period, we also supplement solver hyperparameter tuning results in Sec. 3.4, which indicates that the generated instances maintain close properties to the reference real-world instances so that hyperparameter tuning led by them could bring a significant solver runtime reduction of over 50% to the solver. The evaluation has already shown the effectiveness of MixSAT in generating instances that exhibit the properties intrinsic to real-world instances.
>
> > **Q4: Minor issues and suggestions.**
>
> We sincerely appreciate your thorough review. We will revise the manuscript accordingly to elevate it into a more refined paper.
>
> ---
> We hope this response could help address your concerns. We sincerely wish that you could reconsider the novel contribution and potential impact of our work on the community. We look forward to receiving your valuable feedback soon.
>
> ---
>
> **Reference:**
>
> [1] HardSATGEN: Understanding the Difficulty of Hard SAT Formula Generation and A Strong Structure-Hardness-Aware Baseline. KDD 2023.
>
> [2] G2SAT: Learning to Generate SAT Formulas. NeurIPS 2019.
>
> [3] On the Performance of Deep Generative Models of Realistic SAT Instances. SAT 2022.
>
> [4] Learning latent permutations with gumbel-sinkhorn networks. ICLR 2018.

---

> > ### Comment · Reviewer_5Kyc · 2023-11-21
> > **Reply to the authors 2**
> >
> > Again, thank you for revising the manuscript. My understanding is that the main benefit claimed in the paper now is that it somewhat outperforms the competition in the case of satisfiable instances although it loses to the competition for the unsatisfiable ones. The new discussion should clearly facilitate a reader's understanding of how far/close the generated instances are from/to those they are meant to resemble.

---

> > > ### Author Response · Authors · 2023-11-22
> > > **Response to Reviewer 5Kyc**
> > >
> > > Thanks for your timely reply and the acknowledgment of our efforts in rebuttal. Below we respond to your remaining concerns.
> > >
> > > > **My understanding is that the main benefit claimed in the paper now is that it somewhat outperforms the competition in the case of satisfiable instances although it loses to the competition for the unsatisfiable ones.**
> > >
> > >
> > > This is the case when MixSAT is compared with HardSATGEN in terms of hardness-related evaluations. However, please kindly note the following:
> > >
> > > 1) The primary experiments are beyond mere hardness-related comparisons, encompassing an evaluation of graph structure as well.  In graph structure resemblance performance, MixSAT demonstrates clear superiority over all methods, including HardSATGEN. Additionally, the solver tuning results in Sec. 3.4 highlight that MixSAT's overall performance significantly surpasses all previous baselines, including HardSATGEN.
> > >
> > > 2) Although MixSAT performs weaker than HardSATGEN, it can still outperform other baselines in hardness resemblance performance for unsatisfiable instances.
> > >
> > > 3) HardSATGEN appears to be tailored specifically for unsatisfiable data. Considering the general applicability and relevance to real-world scenarios, which often involve both satisfiable and unsatisfiable data, we contend that MixSAT still represents a breakthrough in reproducing hardness for general methods.
> > >
> > > Therefore, while the performance gains in hardness-related evaluations may stem from satisfiable sources, we maintain that MixSAT still exhibits a general superiority in experiments over other baselines.
> > >
> > >
> > > > **The new discussion should clearly facilitate a reader's understanding of how far/close the generated instances are from/to those they are meant to resemble.**
> > >
> > > Thanks for the note. Our evaluation focuses on assessing resemblance performance across three key aspects:
> > >
> > > 1) Graph structure resemblance (Sec. 3.2). As illustrated in Table 1, MixSAT surpasses previous baselines in capturing graph structure, with the average difference between the generated and reference data across various graph metrics below 5%.
> > >
> > > 2) Computational hardness resemblance (Sec. 3.3). Table 2 shows that MixSAT stands out as the first method with general applicability to be capable of replicating the plausible hardness of reference instances. In comparison to HardSATGEN, MixSAT demonstrates superiority in satisfiable sources and, importantly, exhibits a broader applicability.
> > >
> > > 3) Effectiveness in downstream applications like solver tuning (Sec. 3.4). Table 3 shows that the generated instances by MixSAT yield highly positive results in improving solver performance. This indicates that the generated instances exhibit a higher level resemblance to the reference instances. MixSAT outperforms all other previous baselines in overall performance, bringing a significant solver runtime reduction of over 50% to the solver. Furthermore, in collaboration with HardSATGEN, MixSAT proves even more potent, delivering a performance gain of over 65% to the solver.
> > >
> > > ---
> > >
> > > We hope this response could help address your remaining concerns. We sincerely wish that you could reconsider the novel contribution and potential impact of our work on the community. Your valuable feedback is eagerly anticipated, and we look forward to hearing from you soon.

---

### Official Review · Reviewer_KdVD · 2023-10-31

**Soundness:** 4 excellent
**Presentation:** 3 good
**Contribution:** 3 good
**Rating:** 8
**Confidence:** 3

**Summary:**

This paper proposes MixSAT, a ML-based generation approach for SAT instances. Instead of generating formulas ex novo, MixSAT interpolates exising pairs of formulas using:

-    GNN-based representation learning for capturing structural properties in a latent representation
-    Differentiable Gumbel-Sinkhorn stochastic matching between instances
-    Graph interpolation via iterative clause replacement

The generated formulas statistically retain similar structural properties and preserve the hardness of the original instances.

**Strengths:**

-    Overall very clear presentation
-    Well-motivated problem
-    The proposed technique seems sound and novel
-    Promising initial results

**Weaknesses:**

-    Minor points related to the presentation and evaluation
-    I would also discuss the limitations of MixSAT

**Questions:**

1) Besides structural properties and hardness, does MixSAT preserve the SAT/UNSAT ratio of the original data?
2) What are the limitations of your approach?
3) Do you observe a degratation in performance when interpolating formulas generated by MixSAT?
4) Is it possible to learn a binary classifier that accurately discriminates original vs. MixSAT instances?

Minors:

"The Boolean satisfiability problem (SAT) stands as a canonical NP-complete combinatorial optimization (CO) problem"

SAT is a decision problem, not a CO. Max-SAT is the optimization counterpart.

"there exists an assignment of Boolean variables that satisfies a Boolean formula and in general is NP-hard."

SAT is NP-complete. I understand that, in principle, the formulas that MixSAT generates could be used in non-decision settings too. I would then rephrase the sentences above, making it clear that you address generation of propositional logic instances for decision (SAT) and other problems (e.g. Max-SAT)

"e.g. in EDA."

What EDA stands for?

Adding the pseudocode of MixSAT would greatly help.

---

> ### Author Response · Authors · 2023-11-19
> **Response by Authors (1/2)**
>
> Thanks for the valuable comments, nice suggestions, and for acknowledging the presentation clarity, methodology novelty, and technical soundness of our paper. Below we respond to your specific comments.
>
> > **Q1: Besides structural properties and hardness, does MixSAT preserve the SAT/UNSAT ratio of the original data?**
>
> Thanks for the comment. The results of satisfiability phase preservation for generated formulas are presented below and also in Table 10 of the current revision. It is observed that despite being trained on a dataset with a one-to-one ratio of satisfiable to unsatisfiable instances, previous methods exhibit a tendency to generate a specific satisfiability phase. Notably, instances generated by G2SAT, GCN2S, and HardSATGEN are all unsatisfiable, while PS generates exclusively satisfiable instances. Among all the methods, MixSAT maintains the closest phase ratio to the original reference instances.
>
> Unsatisfiable ratio of the generated instances:
>
> |  **Method**  | **EASY** | **HARD** |
> |:-:|:-:|:-:|
> | Ground Truth | 50.00%  | 50.00%  |
> |  CA  |  87.5%  | 75.00%  |
> |  PS  |  0.00%  | 0.00%  |
> |  G2SAT | 100.00% | 100.00% |
> | GCN2S | 100.00% | 100.00% |
> |  HardSATGEN  |  100%   | 100.00% |
> | MixSAT | 23.61%  | 43.41%  |
>
> > **Q2: What are the limitations of your approach?**
>
> Currently, the primary limitation of our approach lies in the fact that our hardness reproducing performance on unsatisfiable examples, though better than other methods, is not as effective as that achieved by HardSATGEN [1]. However, please note that reproducing computational hardness in the SAT instance generation task is considerably challenging which has not been achieved until [1]. Its success stems from its fine-grained control over the instance structure, particularly for the unsatisfiable core structures, as well as the postprocessing design involved to maintain the dominant effect of the unsat core structure.
>
> While the design in [1] is effective, it confines the method to unsatisfiable instances only. Since real-world datasets generally incorporate both satisfiable and unsatisfiable instances, merely learning from the unsat instances can lead to certain negative impacts on downstream tasks like hyperparameter tuning, as evidenced by Table 3. In contrast, MixSAT surpasses this limitation, demonstrating generality to both satisfiable and unsatisfiable instances. MixSAT is also free from the node merge-split framework of previous learning-based methods [1-3], naturally meeting the hardness analysis and requirements set forth in [1].  Notably, our performance on satisfiable instances is significantly superior to all other methods.
>
> On the other hand, we believe the two methods can complement each other. As evidence, we have supplemented the study with hyperparameter optimization experiments and present the combined effects of both methods in Table 3 of the current revision, which shows the significant effectiveness of our methods in practice as a boosting tool complementary to HardSATGEN.
>
>
> > **Q3: Do you observe a degratation in performance when interpolating formulas generated by MixSAT?**
>
> To evaluate the effect of secondary interpolation on the performance of formulas generated by MixSAT, we select two representative instance pairs and apply MixSAT ($\lambda=0.1,\tau=1$) to obtain their 5% mixing ratio formulas. We then use these formulas as the start formulas and interpolate them with their corresponding goal formulas, resulting in 10% mixing ratio formulas. We compare these formulas with the original formulas, the 5% start formulas, and the 10% formulas obtained by one interpolation. The comparison is shown below and also in Table 9 of the current revision, where we can see that the secondary interpolation does not degrade the performance of the formulas, as they maintain the same level of hardness as the one-interpolated formulas. This demonstrates the robustness of MixSAT in generating hard formulas through interpolation.
>
> |   | **Ground Truth** |  **5%**  | **Interpolated 10%** | **10%**  |
> |:-:|:-:|:-:|:-:|:-:|
> | **hard $b$-$a$** | 1382.71 | 1023.25 $\pm$ 4.69  | 1001.97 $\pm$ 7.28 | 1042.53 $\pm$ 4.78 |
> | **hard $d$-$c$** | 852.85 | 199.63 $\pm$ 190.70 | 6.13 $\pm$ 3.14 | 3.56 $\pm$ 1.69  |

---

> ### Author Response · Authors · 2023-11-19
> **Response by Authors (2/2)**
>
> > **Q4: Is it possible to learn a binary classifier that accurately discriminates original vs. MixSAT instances?**
>
> Due to the limited data availability in specific SAT scenarios (which is also the motivation for designing generation algorithms), the current dataset size is insufficient to support the training of a classification network. In fact, we believe this situation is widely acknowledged in literature [1,2].
>
> We understand your concern, as classification is a method to discern whether the generated distribution approximates the real one. Indeed, in our experiments, out of the same consideration, we have already validated the structural and hardness similarities between generated and real instances. Additionally, during the rebuttal period, we have highlighted MixSAT's superior performance in solver tuning. We believe these results sufficiently support the approximation of the distribution between the data generated by our method and real-world data.
>
> > **Q5: What EDA stands for?**
>
> EDA stands for "Electronic Design Automation", a practical application scenario of SAT problem, where SAT instances exhibit scarcity. Sorry for the confusion and we have fixed it in the revision.
>
> > **Q6: Pseudocode of MixSAT.**
>
> Thanks for the comment. We supplement the pseudocode in Appendix C to further enhance our methodology presentation clarity.
>
>
>
> > **Q7: Minor issues and suggestions.**
>
> We sincerely appreciate your thorough review. We will revise the manuscript accordingly to elevate it into a more refined paper.
>
>
> ---
> We hope this response could help ease your concern and wish to receive your further feedback soon.
>
> ---
>
> **Reference:**
>
> [1] HardSATGEN: Understanding the Difficulty of Hard SAT Formula Generation and A Strong Structure-Hardness-Aware Baseline. KDD 2023.
>
> [2] G2SAT: Learning to Generate SAT Formulas. NeurIPS 2019.
>
> [3] On the Performance of Deep Generative Models of Realistic SAT Instances. SAT 2022.
>
> [4] Locality in random SAT instances. IJCAI 2017.

---

### Author Response · Authors · 2023-11-19
**General Response by Authors**

Dear Area Chairs and Reviewers,

We greatly appreciate the reviewers' time, valuable comments, and constructive suggestions. Overall, the reviewers deem our idea and methodology "sound" (KdVD), "novel" (KdVD, 5Kyc), and "interesting and carefully designed" (SBgK), acknowledging our experiments (KdVD, SBgK), with "very clear", "easy to read" or "nice" presentation (KdVD, 5Kyc, SBgK), and "beautiful and helpful figures" (5Kyc, SBgK).


In the author response period, we make every effort to address reviewers’ concerns and provide additional experimental results to further verify our contributions. The summary of our main efforts corresponding to the new revision is presented as follows:

1) We provide solver hyperparameter tuning application results in Table 3. We use the generated instances to tune the SOTA  solver Kissat to observe whether the augmented data could directly benefit the solver's performance. In general, the generated instances by our proposed methods bring a significant solver efficiency gain of over 50\% to the solver. See details in Sec. 3.4.
2) We augment the ablation study by examining the impact of introducing randomness for instances experiencing a degradation in hardness. Table 5 illustrates the influence of introducing randomness to alleviate the hardness degradation issue. See details in Sec. 3.5.
3) We emphasize the main motivation or rationale behind the specific designs of our approach in blue in the uploaded revision. This is to further enhance the presentation clarity of our methodology.
4) We supplement solver runtime evaluation on more mainstream solvers to further validate our performance on hardness resemblance in Tables 6 and 7.
5) We compare MixSAT's runtime to the previous node split-merge framework in Table 8 to show the significant efficiency advantage of MixSAT.
6) We supplement the results of satisfiability phase preservation in Table 10 and show that previous methods typically exhibit an extreme tendency to generate a specific satisfiability phase, while MixSAT maintains the closest phase ratio to the original reference instances.
7) We supplement the pseudocode in Appendix C to further enhance our methodology presentation clarity.


In our individual responses, we provide detailed answers to all the specific questions raised by the reviewers. We hope these responses could help address the reviewers' concerns, and further discussions are welcomed toward a comprehensive evaluation of our work.

---

### Author Response · Authors · 2023-11-20
**Sincerely Awaiting Your Feedback**

Dear Reviewers,

We would like to express our sincere gratitude again for your valuable comments and thoughtful suggestions. Throughout the rebuttal phase, we tried our best to address concerns, augment experiments to fortify the paper (comprising approximately 4 pages of new content with 7 new tables), and refine details in alignment with your constructive feedback. Since the discussion time window is very tight and is approaching its end, we truly hope that our responses have met your expectations and assuaged any concerns. We genuinely do not want to miss this opportunity to engage in further discussions with you, which we hope could contribute to a more comprehensive evaluation of our work. Should any lingering questions persist, we are more than willing to offer any necessary clarifications.

With heartfelt gratitude and warmest regards,

The Authors

---

### Author Response · Authors · 2023-11-23
**Rebuttal Summary and Kindly Awaiting Further Feedback**

Dear Reviewers and ACs,

We sincerely appreciate your valuable engagement and the time you dedicated to reviewing our work. Throughout the author-reviewer discussion period, the reviewers acknowledged our efforts in the detailed responses and in revising the paper. While new questions were raised concerning the resemblance degree between the generated and reference data (5Kyc), details about the new results including Tables 3 and 8 (SBgK), and specifics about dataset construction (SBgK). We have diligently addressed the remaining concerns as follows:

1) We provided details about how we measure the data resemblance (from three aspects compromising structure, hardness, and solver tuning results), and how well MixSAT performs regarding the resemblance compared to previous baselines.

2) We provided more details of Table 3 and Table 8, offering justifications for the specific settings and explaining how they positively contribute to the evaluation of SAT generation methods. These supplementary results also highlight MixSAT's superiority over previous baselines, emphasizing its contribution to the community.

3) To address concerns about dataset construction, we have provided specific details, underscoring the close adherence of the data to G2SAT. We have also uploaded our code at https://anonymous.4open.science/r/MixSAT/ to ease the reviewer's concern, which can be evaluated by any data to prove MixSAT's performance.

As the discussion phase nears its conclusion, we are still wondering whether our general rebuttal has effectively met the reviewers' requests and if the concerns raised in the initial reviews and subsequent discussions have been satisfactorily mitigated. We would sincerely appreciate it if you could reconsider the overall contribution and the potential impact of our work based on the current version of our paper, which we believe is largely strengthened during this tight rebuttal phase. Your feedback at this juncture would be highly valuable to us.

With heartfelt gratitude and warmest regards,

The Authors

---

### Meta-Review · Area_Chair_aBik · 2023-12-07

**Metareview:**

Summary:
This paper is devoted to the machine learning-assisted generation of families of benchmark formulas for propositional satisfiability (SAT) solvers. The paper builds on graph representations of formulas in conjunctive normal form (CNF). It utilizes a graph interpolation approach, modifying the graph structure by replacing some of its substructures from similar instances. The paper argues that the proposed method can maintain the original hardness of the corresponding families of benchmarks, thanks to Gumpel noise. The presented experimental results claim to demonstrate the advantage over the state of the art in SAT benchmark generation from the perspective of (1) maintaining the structure of the benchmark family and (2) its hardness.

Strengths:
+ Overall, obvious presentation
+ Well-motivated problem
+ The proposed technique seems sound and novel
+ Promising initial results

Weaknesses:
- The steps of the approach are not augmented with clear arguments for why they are used.
- Experimental results look relatively weak in places.
- Some crucial details of experiments are missing
- Some claims not supported by experiments

**Justification For Why Not Higher Score:**

It is not a clear accept that would make attendees learn in more detail about this work. However, there will be a subgroup of researchers in ICLR that would find this work interesting

**Justification For Why Not Lower Score:**

For the reasons above.

---

### Decision · Program_Chairs · 2024-01-16

Accept (poster)